# MreC and MreD balance the interaction between the elongasome proteins PBP2 and RodA

Xiaolong Liu[1¤], Jacob Biboy[2], Elisa Consoli[1], Waldemar Vollmer[2], Tanneke den Blaauwen[1]*

**1** Bacterial Cell Biology & Physiology, Swammerdam Institute for Life Science, Faculty of Science, University of Amsterdam, Amsterdam, The Netherlands, **2** Centre for Bacterial Cell Biology, Biosciences Institute, Newcastle University, Newcastle upon Tyne, United Kingdom

¤ Current address: Department of Biochemistry, University of Oxford, Oxford, United Kingdom
* t.denblaauwen@uva.nl

**Data Availability Statement:** All relevant data are within the manuscript and its Supporting Information files.

**Funding:** X.L. was supported by the Chinese Scholarship Council (File No.201406220123).

## Abstract

Rod-shape of most bacteria is maintained by the elongasome, which mediates the synthesis and insertion of peptidoglycan into the cylindrical part of the cell wall. The elongasome contains several essential proteins, such as RodA, PBP2, and the MreBCD proteins, but how its activities are regulated remains poorly understood. Using *E. coli* as a model system, we investigated the interactions between core elongasome proteins *in vivo*. Our results show that PBP2 and RodA form a complex mediated by their transmembrane and periplasmic parts and independent of their catalytic activity. MreC and MreD also interact directly with PBP2. MreC elicits a change in the interaction between PBP2 and RodA, which is suppressed by MreD. The cytoplasmic domain of PBP2 is required for this suppression. We hypothesize that the *in vivo* measured PBP2-RodA interaction change induced by MreC corresponds to the conformational change in PBP2 as observed in the MreC-PBP2 crystal structure, which was suggested to be the "on state" of PBP2. Our results indicate that the balance between MreC and MreD determines the activity of PBP2, which could open new strategies for antibiotic drug development.

## Author summary

The cell envelope of *Escherichia coli* bears the protective and shape-determining peptidoglycan layer sandwiched between the outer and inner membranes. Length growth in bacteria is accomplished by a protein complex termed elongasome. In this complex, RodA and PBP2 provide the peptidoglycan glycosyltransferase and transpeptidase activities needed to synthesize new peptidoglycan during length growth, respectively, and PBP2 activates RodA. Using Förster Resonance Energy Transfer (FRET) that reports not only on whether proteins interact with each other but also on conformational changes during interactions, we investigated how RodA and PBP2 interact. We show that the interactions between MreC and MreD with PBP2-RodA alter the nature of the interaction between PBP2 and RodA and hypothesize that the corresponding conformational change in the

https://www.chinesescholarshipcouncil.com/ W.V.
received funding from the Wellcome Trust
(101824/Z/13/Z). https://wellcome.ac.uk/?gclid=
Cj0KCQiA-

PBP2-RodA complex allows switching between the 'on' and 'off' states of PBP2 activity,
contributing to the regulation of elongasome activity.

## Introduction

Bacterial cells are surrounded by a peptidoglycan layer that maintains their shape and protects
them from bursting due to the osmotic pressure. The biosynthesis of peptidoglycan is the tar-
get of many antibiotics that are used in clinical therapies for bacterial infections. The spread of
antibiotic resistant pathogens calls urgently for the development of novel antibiotics. In depth
knowledge on peptidoglycan synthesis will aid in the development of effective screening assays
to select cell wall synthesis inhibitors. Peptidoglycan is a mesh-like heteropolymer of glycan
chains of *N*-acetylglucosamine-*N*-acetylmuramoyl-peptides (Glc*N*Ac-Mur*N*Ac-peptide) sub-
units that are connected by peptide cross-links [1]. Peptidoglycan synthesis begins in the cyto-
plasm with synthesis of UDP-Glc*N*Ac and UDP-Mur*N*Ac-pentapeptide [2]. Two following
membrane steps, catalyzed by MraY and MurG, assemble the precursor lipid II [3, 4], which is
flipped to the periplasmic side of the cytoplasmic membrane by lipid II flippase(s) MurJ and/
or FtsW [5–8]. Glc*N*Ac-Mur*N*Ac pentapeptide units are polymerized into glycan chains and
the peptides are cross-linked to bridge the glycan stands by peptidoglycan synthases to expand
the peptidoglycan layer while the carrier lipid is recycled [9–11]. Most rod-shaped bacteria
employ two protein complexes, elongasome and divisome, to guide peptidoglycan synthesis
during lateral growth and cell division, respectively [12].

In *E. coli*, the divisome contains more than twenty proteins. Assembly of the divisome starts
with positioning the FtsZ ring at midcell together with other early divisome proteins, such as
FtsA, ZipA, ZapA and FtsEX, to form the early divisome [13–15]. Subsequently, the late divi-
some proteins, FtsK, FtsBLQ, FtsW, PBP3 and FtsN, are recruited [16]. These proteins localize
to midcell in an interdependent order [16, 17]. Among these proteins, FtsW, PBP3 and PBP1B
provide the peptidoglycan synthesis activity during septum synthesis [11, 18, 19]. PBP1B has
both glycosyltransferase (GTase) and transpeptidase (TPase) activity [20], while FtsW and
PBP3 (also called FtsI) only have GTase activity and TPase activity, respectively [11]. Although
the mechanisms of peptidoglycan synthesis regulation is not fully understood, recent studies
showed that the cell division proteins have competing effects and either inhibit (FtsQLB com-
plex) or stimulate the activities of FtsW-PBP3-PBP1B [21–23].

Proteins that are known to be part of the elongasome are the cytoplasmic membrane associ-
ated actin homologue MreB, the bitopic membrane proteins RodZ, MreC and PBP2, and the
integral membrane proteins MreD and RodA (Fig 1A). MreB polymerizes into short filaments
that rotate around the cylindrical part of the cell [24, 25]. The rotation of MreB is believed to
drive the topography of the insertion of peptidoglycan into the lateral wall [24, 26–28]. Bacte-
rial two hybrid analysis showed that MreB interacts with MreC, but not with MreD [29], while
RodZ interacts strongly with itself, MreB and MreC [29, 30] (Fig 1B), and these interactions
are essential to maintain bacterial morphology [29, 31–33]. RodA and PBP2 form a stable sub-
complex [34, 35] and provide GTase and TPase activity, respectively, during cylindrical pepti-
doglycan synthesis [10, 36, 37]. This subcomplex also shows a circumferential motion that is
similar to that of MreB. The bifunctional GTase-TPase PBP1A interacts with PBP2 and stimu-
lates its activity [19]. Because PBP1A moves independently of the rotation of PBP2 and MreB,
it is thought not to be part of the core elongasome [19, 38]. A recent study revealed that the
*Helicobacter pylori* PBP2 has two different structural conformations when in complex with or
in the absence of MreC, which were proposed to correspond to the "on" (active) and "off"

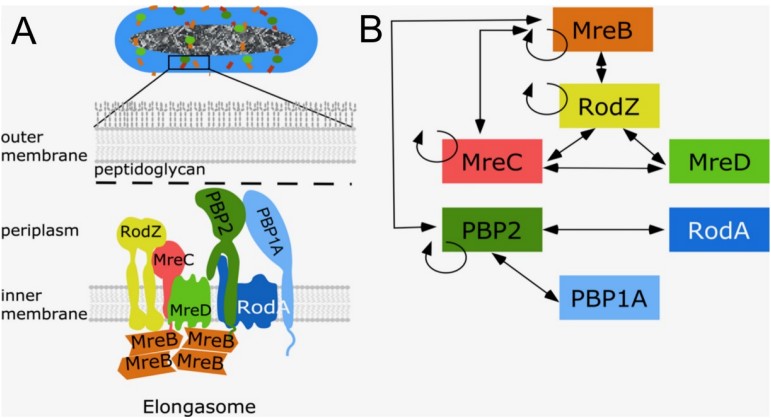

**Fig 1. Core elongasome proteins and their interactions in *E. coli*. a.** Schematic representation of the *E. coli* cell envelope and elongasome. MreB localizes in patches underneath the cytoplasmic membrane and recruits other elongasome proteins. The peptidoglycan layer is sandwiched by the cytoplasmic membrane and the outer membrane. **b**. Identified interactions between elongasome proteins from previous interaction studies [18, 19, 29, 34]. Double arrowed lines represent the interaction between different proteins. Circular arrows indicate self-interaction.

(inactive) state of PBP2, respectively [39]. In *E. coli*, a hyperactive PBP2$^{L61R}$ was identified to partially suppress the MreC defects, and was proposed to stay at the active state [40]. However, the function and role of most elongasome proteins, and how peptidoglycan synthesis is activated and regulated during elongation, is still poorly understood. In this study, combining genetics, microscopy and Förster Resonance Energy Transfer (FRET), we investigated the functions of, and interactions between, these core elongasome proteins. The transfer of energy between a donor fluorescent-protein fusion and an acceptor fluorescent-protein fusion (FRET) is very sensitive to distance, which even allows the detection of conformational changes that affect this distance [41]. Our results indicate that MreC and MreD modulate the interaction between PBP2 and RodA oppositely, which likely reflects a mechanism of elongasome activation and regulation.

## Results

### RodA and PBP2 activities are not essential for their interaction

RodA and PBP2 form a stable peptidoglycan synthesizing subcomplex in the cytoplasmic membrane as detected by FRET [34]. To investigate whether this interaction relies on their enzymatic activities, RodA$^{R109A}$ and RodA$^{Q207R}$ versions, which were predicted to be inactive based on studies on its homologue FtsW, were constructed (S1 Fig) [5, 42]. As expected, these mutants could not complement the temperature sensitive RodA strain LMC882 at the non-permissive temperature, and the RodA$^{Q207R}$ variant even showed dominant negative effects at the permissive temperature (Fig 2A). Subsequently, N-terminal mCherry fused versions [34, 36] of the inactive RodA proteins were expressed to test their interaction with mKO-PBP2$^{WT}$ by FRET (Fig 2B). In our FRET system, the direct fused mCherry-mKO tandem was used as positive control [34]. To account for possible interactions between proteins due to crowding in the cytoplasmic membrane, an integral membrane protein unrelated to peptidoglycan synthesis, GlpT, was fused to mKO, and its interaction with mCh-RodA was detected as negative control. The acceptor FRET efficiency values (Ef$_A$) of all FRET samples were calculated using our previously published mKO-mCh FRET spectral unmixing method [34] (S2A Fig). The unmixed spectra reveal the intensity of the mKO and mCherry, which allows verification of the amount of protein expressed by the cells (Supplementary figures contain all unmixed

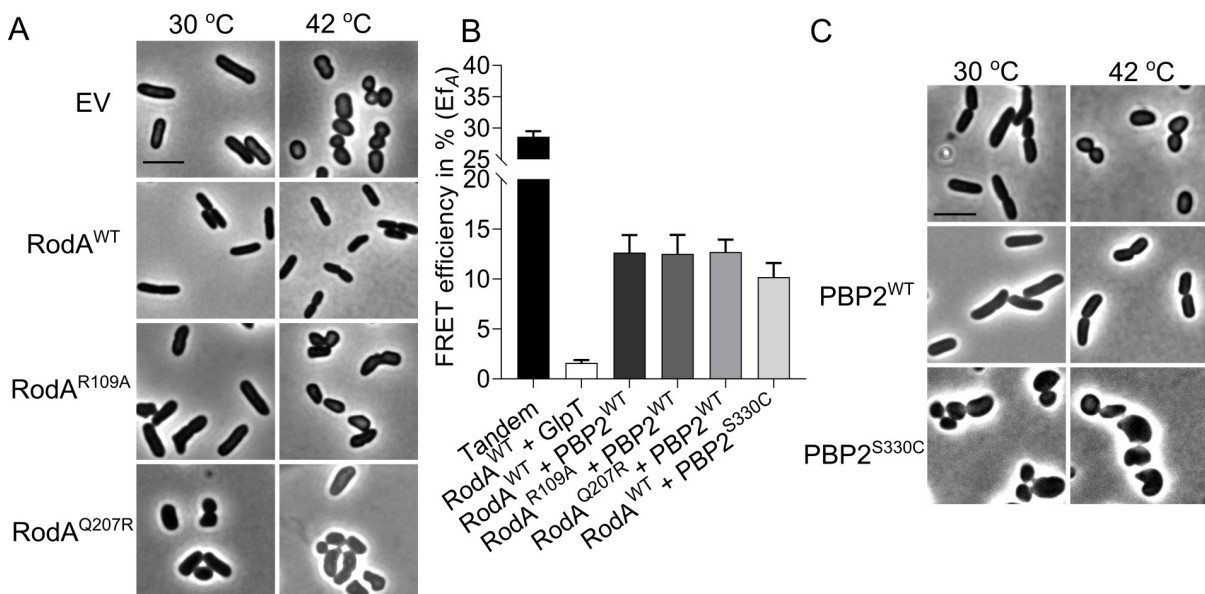

**Fig 2. Activity of RodA and PBP2 are not required for their interaction. a**, Phase contrast images of the complementation of RodA variants. RodA temperature sensitive strain LMC882 was transformed with plasmids expressing RodA variants and grown in LB medium at 30°C (left panel) and 42°C (right panel) for 2 mass doublings (with15 μM IPTG induction). EV, empty vector. **b**, Calculated acceptor FRET efficiencies ($Ef_A$) between PBP2 and RodA variants from spectral FRET measurements. RodA and its variants are fused with mCherry. PBP2 and its variants are fused with mKO. **c**. Phase contrast images of the complementation of PBP2 variants. The PBP2 temperature sensitive strain LMC582 was transformed with $PBP2^{WT}$ or $PBP2^{S330C}$, and grown in LB medium at 30°C (left panel) and 42°C (right panel) for 2 mass doublings (with 15 μM IPTG induction). Scale bar equals 5 μm. All the results in the figure are representative of at least three independent experiments.

spectra). Expression levels for the various constructs were similar. $Ef_A$ value of $31.0 \pm 4.0\%$ was observed for the tandem control (Fig 2B and Table 1), which is comparable to the published data [34]. An $Ef_A$ value of $1.1 \pm 3.5\%$ was observed for the RodA+GlpT negative control (Fig 2B and Table 1). FRET experiments of $PBP2^{WT}+RodA^{R109A}$ or $RodA^{Q207R}$ yielded $Ef_A$ values of $12.5 \pm 1.9\%$ and $12.7 \pm 1.2\%$, respectively, which are comparable to the $Ef_A$ value of $12.7 \pm 1.7\%$ of wild-type RodA, indicating an unchanged interaction between $PBP2^{WT}$ and all RodA versions (Figs 2B and S2A and Table 1). Similarly, to determine whether the activity of PBP2 was required for the interaction with RodA, we expressed the inactive variant $PBP2^{S330C}$, which is not able to bind benzylpencillin covalently [37]. $PBP2^{S330C}$ had a strong dominant negative effect during the complementation in the PBP2 temperature sensitive strain LMC582 (Fig 2C), while the detected $Ef_A$ value of $PBP2^{S330C}+RodA^{WT}$ was with $10.2 \pm 1.4\%$ not much different from the $Ef_A$ value of $PBP2^{WT}+RodA^{WT}$ (Fig 2B and Table 1). These results imply that the activities of RodA and PBP2 are not needed for their interaction.

## The transmembrane and periplasmic parts of PBP2 contribute to its interaction with RodA

To reveal which part of PBP2 interacts with RodA, two domain swap mutants of PBP2 were constructed. The cytoplasmic N-terminus (NT) and or the N-terminal region with the trans-membrane-helix (NT-TMH) of PBP2 were replaced by the corresponding N-terminal stretches of MalF, a bitopic membrane protein as shown in domain swap studies [43–45], to yield $^{MalFNT}$PBP2 and $^{MalF37}$PBP2, respectively (Fig 3A). Both versions of PBP2 were able to localize to the membrane but showed dominant negative effects, indicating the essentiality of the replaced parts (Fig 3B). The replacement of the NT of PBP2 did not change its interaction

**Table 1. Summary of the calculated acceptor FRET efficiencies (Ef$_A$) from spectral FRET measurements for listed samples.**

| Parameter | Proteins expressed | | Ef$_A$ (%) | SD (%) | N[1] |
| --- | --- | --- | --- | --- | --- |
| | pTHV037 | pSAV057 | | | |
| **Positive control[2]** | | | | | |
| Tandem | Empty plasmid | mKO-mCh | 31.0 | 4.0 | 22 |
| **Negative control[3]** | | | | | |
| RodA-GlpT | mCh-RodA | mKO-(GGS)$_2$-GlpT | 1.1 | 3.5 | 24 |
| **Biological interactions** | | | | | |
| RodA[4]+PBP2 $^{WT}$ | mCh-RodA | mKO-PBP2$^{WT}$ | 12.7 | 1.7 | 16 |
| RodA$^{R109A}$+PBP2 | mCh-RodA$^{R109A}$ | mKO-PBP2 $^{WT}$ | 12.5 | 1.9 | 4 |
| RodA$^{Q207R}$+PBP2 | mCh-RodA$^{Q207R}$ | mKO-PBP2 $^{WT}$ | 12.7 | 1.2 | 4 |
| RodA+PBP2$^{S330C}$ | mCh-RodA | mKO-PBP2$^{S330C}$ | 10.2 | 1.4 | 4 |
| RodA+$^{MalFNT}$PBP2 | mCh-RodA | mKO-$^{MalFNT}$PBP2 | 11.8 | 4.1 | 6 |
| RodA+$^{MalF37}$PBP2 | mCh-RodA | mKO-$^{MalF37}$PBP2 | 8.2 | 1.3 | 3 |
| RodA+PBP2$^{L61R}$ | mCh-RodA | mKO-PBP2$^{L61R}$ | 10.8 | 3.0 | 8 |
| RodA+RodA$^{WT}$ | mCh-RodA | mKO-RodA | 5.5 | 1.4 | 4 |
| MreB$^{SW}$+RodA | mCh-MreB$^{SW5}$ | mKO-RodA | 7.2 | 3.4 | 4 |
| PBP2 $^{WT}$ +PBP2 $^{WT}$ | mCh-PBP2 $^{WT}$ | mKO-PBP2 | 9.2 | 0.5 | 4 |
| PBP2+$^{MalFNT}$PBP2 | mCh-PBP2 $^{WT}$ | mKO-$^{MalFNT}$PBP2 | 10.5 | 2.0 | 4 |
| MreC+PBP2 $^{WT}$ | mCh-MreC | mKO-PBP2 $^{WT}$ | 5.1 | 1.2 | 6 |
| MreC+$^{MalFNT}$PBP2 | mCh-MreC | mKO-$^{MalFNT}$PBP2 | 5.4 | 1.7 | 6 |
| MreC+PBP2$^{L61R}$ | mCh-MreC | mKO-PBP2$^{L61R}$ | 5.3 | 0.6 | 2 |
| MreD+PBP2 $^{WT}$ | mCh-MreD | mKO-PBP2 $^{WT}$ | 4.3 | 1.2 | 10 |
| MreD+$^{MalFNT}$PBP2 | mCh-MreD | mKO-$^{MalFNT}$PBP2 | 5.3 | 1.6 | 10 |
| MreCD[6]+PBP2 $^{WT}$ | mCh-MreCD | mKO-PBP2 $^{WT}$ | 3.3 | 0.5 | 6 |
| MreCD[7]+$^{MalFNT}$PBP2 | mCh-MreCD | mKO-$^{MalFNT}$PBP2 | 5.4 | 1.1 | 5 |
| **+ mecillinam** | | | | | |
| **Positive control** | | | | | |
| Tandem | Empty vector | mKO-mCh | 30.3 | 2.7 | 4 |
| **Negative control** | | | | | |
| RodA+GlpT | mCh-RodA | mKO-(GGS)$_2$-GlpT | 1.0 | 1.6 | 4 |
| **Biological interactions** | | | | | |
| RodA+PBP2 $^{WT}$ | mCh-RodA | mKO-PBP2 $^{WT}$ | 7.9 | 1.8 | 8 |
| RodA+$^{MalFNT}$PBP2 | mCh-RodA | mKO-$^{MalFNT}$PBP2 | 6.2 | 1.1 | 4 |
| RodA+$^{MalF37}$PBP2 | mCh-RodA | mKO- $^{MalF37}$PBP2 | 3.0 | 2.0 | 4 |
| RodA+PBP2$^{S330C}$ | mCh-RodA | mKO-PBP2$^{S330C}$ | 6.6 | 1.7 | 4 |
| RodA+PBP2$^{L61R}$ | mCh-RodA | mKO-PBP2$^{L61R}$ | 6.3 | 3.5 | 7 |

[1] No, number of biological repeats. See for all individual values S1 Data.

[2,3] Here all measured positive and negative controls are averaged. In the figures the controls are included that belong to the corresponding measurements.

[4] RodA without superscript represent the wild-type version.

[5] mCh-MreB$^{SW}$ is a sandwich fusion of MreB-mCherry-MreB [33].

[6,7] MreC and MreD were expressed from one plasmid, and MreC was fused to mCherry while MreD was non-fused.

$p$ values for comparisons are shown in the figures.

with RodA, as the detected Ef$_A$ value remained 11.8 ± 4.1%, which was not significantly different compared to that of the interaction between RodA and wild-type PBP2 (Figs 3C and S2B and Table 1). However, replacement of the TMH of PBP2 significantly reduced the Ef$_A$ value between PBP2 and RodA to 7.9 ± 1.8%, which reflected an apparent distance increase from 8.8 nm to 9.6 nm between the two proteins [46] (Fig 3C and Table 1). This increase in distance

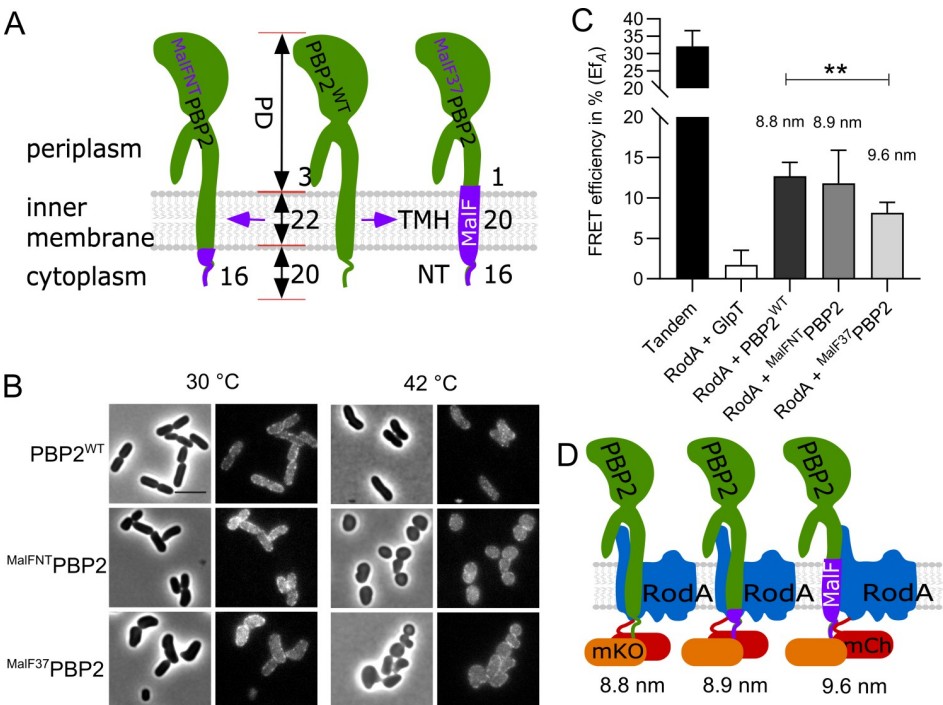

**Fig 3. Functionality and interaction of PBP2 domain swap mutants. a.** Schematic illustration of PBP2 domain-swap mutants. NT: N-terminus; TMH: transmembrane helix; PD: periplasmic domain; PBP2^WT: wild-type PBP2; ^MalFNT^PBP2: the cytoplasmic N-terminus of PBP2 was replaced with the MalF cytoplasmic N-terminus; ^MalF37^PBP2: the NT and TMH domains of PBP2 were replaced with corresponding domains of MalF. Numbers indicate the residues involved in replacements in each domain of the proteins. **b.** Phase contrast and fluorescence images of the complementation of PBP2 mutants. The PBP2 temperature sensitive strain LMC582 was transformed with the PBP2 variants, and grown in LB medium at 30˚C (left panels) and 42˚C (right panels) for 2 mass doublings (with 15 μM IPTG induction). Scale bar equals 5 μm. **c.** Calculated acceptor FRET efficiencies (Ef$_A$) between PBP2 and RodA variants from spectral FRET measurements. RodA and its variants are fused with mCherry. PBP2 and its variants are fused with mKO. P value determined with Student's t-test (**: p<0.001). The numbers are apparent distances between the two proteins (fluorophores) and were calculated using the equation, $E = (1+(r/R_0)^6)^{-1}$, where r is the distance between the chromophores and R$_0$ (the Förster distance) is 6.4 nm for the mCh-mKO pair. **d.** Schematic illustration of the interaction between RodA and PBP2 variants. After replacement of the TMH domain of PBP2, by the TMH of MalF, the FRET efficiency is lower indicating that the distance between the FP fused TMHs of PBP2 and RodA has increased. However the interaction is not lost, indicating that periplasmic domains are also involved in the interaction All results in the figure are representative of at least three independent experiments.

was not caused by a change in amount of measured fluorescence for the RodA and PBP2 fusions expressed in the cells (S2B Fig), or due to the shorter transmembrane helix after replacement (Fig 3A). The average rise per residue in transmembrane helices is 0.15 nm [47], therefore the two amino acid residues shorter helix in ^MalF37^PBP2 (Fig 3A) could maximally change the distance between donor and acceptor fluorophores by 0.3 nm. The still considerably higher Ef$_A$ value compared to the negative control indicates that the transmembrane helix alone is not sufficient for the wild-type interaction between PBP2 and RodA and that the periplasmic domain of PBP2 is also involved in this interaction (Fig 3D) as was recently confirmed by the crystal structure of the *Thermus thermophilus* RodA-PBP2 complex [48].

## MreC interacts with PBP2 and affects PBP2-RodA interaction

A recent study of PBP2-MreC from *Helicobacter pylori* showed two different structural conformations of PBP2 in the MreC-bound and unbound forms [39] (Fig 4A). The authors proposed that the binding of MreC to the periplasmic hydrophobic zipper of PBP2 induces a

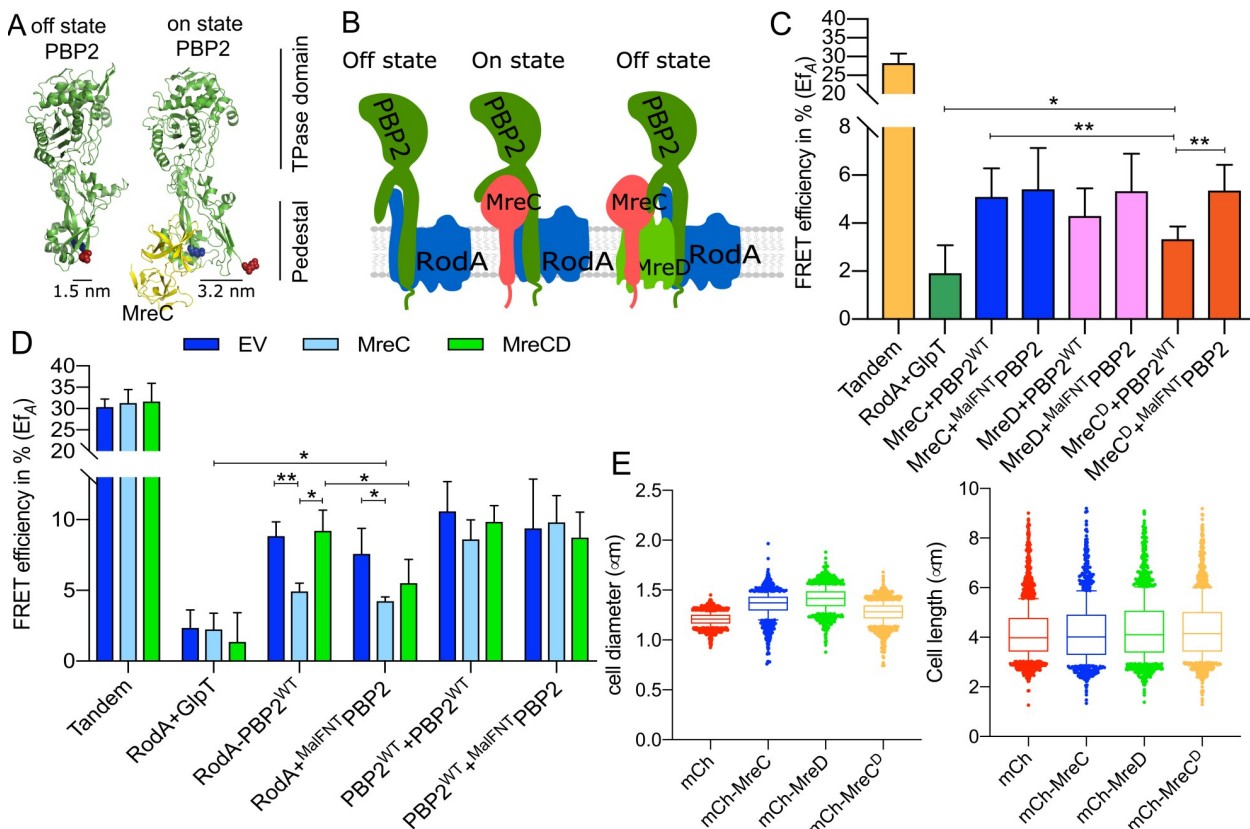

**Fig 4. The balance of MreC and MreD affects the interaction between RodA and PBP2. a**. Crystal structures of *E. coli* PBP2 in different conformations [40] were modeled from *Helicobacter pylori* PBP2 structures [39] using Phyre2 [68]. The structural information lacks the juxta-membrane, transmembrane helix and cytoplasmic regions of PBP2. MreC binds to PBP2 and was proposed to switch PBP2 from the "off state" to the "on state" [39]. The distances between the two sphered residues, (blue for Glu157) and (red for Lys60) were calculated based on the structure of PBP2 in the different conformations. **b**. Schematic representation of PBP2 conformational changes caused by MreC. Left panels: PBP2 stays in the "off state" in the absence of MreC (the distance between the cytoplasmic terminus of RodA and PBP2 is small); middle panels: PBP2 switches to the "on state" after binding MreC (the distance between cytoplasmic terminus of RodA and PBP2 is larger), right panels: MreD suppresses the MreC-mediated conformational change of PBP2 and keeps PBP2 in the "off state". **c**. Calculated acceptor FRET efficiencies ($Ef_A$) between MreCD proteins and RodA variants from spectral FRET measurements. MreC: mCherry fused MreC; MreD: mCherry fused MreD; $MreC^D$: MreCD co-expressed from the same plasmid, and MreC is fused with mCherry while MreD is non-fused. PBP2 and its variants are fused with mKO. **d**. Calculated acceptor FRET efficiencies ($Ef_A$) between RodA and PBP2 variants from spectral FRET measurements in the three-plasmids FRET experiments. EV: a third empty vector; MreC: a third plasmid expressing non-fused MreC; MreCD: a third plasmid expressing non-fused MreCD. **e**. Cell length and diameter changes after expressing MreC, MreD or MreCD together. LMC500 strain was transformed with each construct and grown in LB medium at 37˚C and induced with 15 μM IPTG for 2 mass doublings. Proteins were expressed from the pSAV057 derived plasmids (mKO: control; MreC: mKO fused MreC; MreD: mKO fused MreD; $MreC^D$: co-expression of MreC and MreD, MreC is fused with mKO while MreD is not fused. About 1000 cells were analyzed. P value determined with Student's t-test (*: $p<0.05$; **: $p<0.01$:).

conformational change in PBP2 and a switch from an off state into an on state. In our FRET system about 2000 copies of the mKO fusion proteins per cell are expressed from a plasmid [49]. The ~180 endogenous copies of MreC molecules [50] are not sufficient to activate the majority of the plasmid expressed mKO-PBP2 molecules. Therefore, we hypothesize that most of the mildly overexpressed PBP2 versions remain in the off state conformation (Fig 4B, left). We reasoned that the interaction between PBP2 and RodA could be sensitive to possible conformational changes of PBP2, if we would additionally express MreC to balance the molecule numbers of both proteins. To this end we first tested the interaction between a functional mCh-MreC [29] fusion and mKO-PBP2 by FRET measurements. The observed $Ef_A$ value of 5.1 ± 1.2% indicates a direct interaction between PBP2 and MreC (Fig 4C and S3 and Table 1), which is in agreement with the structural study of PBP2-MreC from *Helicobacter pylori* [39].

We next employed a three-plasmids-FRET system that expressed MreC from a third plasmid when testing the interaction between PBP2 and RodA (Fig 4D and S4 and Table 2). A control strain contained an empty plasmid instead of the MreC-expression plasmid. In the presence of the empty plasmid, the calculated $Ef_A$ values for the tandem (positive control) and RodA+GlpT (negative control) were 30.4 ± 1.8% and 2.3 ± 1.3%, respectively (Fig 4D and Table 2). These $Ef_A$ values remained unchanged in the presence of MreC expressed from the third plasmid (Fig 4D and Table 2). Interestingly, the $Ef_A$ value for the RodA+PBP2 interaction was significantly reduced to 4.9 ± 0.6% in the presence of MreC, compared with the $Ef_A$ of 8.8 ± 1.1% in the presence of empty plasmid (Fig 4D and Table 2). These results indicate that MreC changes the interaction between PBP2 and RodA, which would be consistent with an conformational change of PBP2 from the off state to the on state proposed from the crystal structures [39] (Fig 4D, middle).

## MreD suppresses the MreC-mediated change in the PBP2-RodA interaction

During our study, we noticed that overexpression of MreC caused morphological defects of the wild-type strain grown in rich medium, decreasing the length and increasing the diameter of *E. coli* cells (Figs 4E and S5). Interestingly, the co-expression of MreD together with MreC suppressed at least the decrease in length (Figs 4E and S5). To further investigate this effect, we aimed to express an N-terminal functional mCherry fusion of MreD [31] which is an integral membrane protein with 6 predicted transmembrane helices and both termini localized in the cytoplasm (S6 Fig). Consistent with this topology model we readily observed fluorescence signals for N-terminal fused GFP-MreD [31] and mKO-MreD versions, which could not be possible if the N-terminus of MreD would localize in the oxidative periplasm where GFP and mKO do not mature [41]. Similarly as for MreC, the overexpression of MreD alone resulted in morphological defects of *E. coli* cells (Figs 4E and S5 and S2 Data). These morphological defects were also observed when expressing untagged MreC and MreD proteins from plasmid in rich medium at 37˚C. In contrast to cells growing in minimal medium at 28˚C that hardly showed any morphological changes, which can be likely attributed to the lower protein expression level (Figs 5C and S5B).

To study the role of MreD in the elongasome, FRET experiments were designed to detect a possible interaction with PBP2. The $Ef_A$ value of 4.3 ± 1.1% indicated a direct interaction between MreD and PBP2 (Figs 4C and S3 and Table 1). Subsequently, the interaction between MreC and PBP2 was measured by FRET in the presence of MreD. The calculated $Ef_A$ between MreC and PBP2 was significantly reduced from 5.1 ± 1.2% to 3.3 ± 0.5% (p = 0.0078) when MreD was co-expressed (Figs 4C and S3 and Table 1). Since MreC reduced the $Ef_A$ of RodA +PBP2 from 8.8 ± 1.1% to 4.9 ± 0.5%, it was possible that MreD altered the effect of MreC on the RodA-PBP2 interaction. Therefore, the three-plasmid FRET experiment was applied to detect the interaction between RodA and PBP2 in the presence of MreCD. Interestingly, the $Ef_A$ value of RodA+PBP2 was restored to 9.2 ± 1.5%, which was comparable with the $Ef_A$ in the presence of the third, empty plasmid (Fig 4D and Table 2). Expression of MreD without MreC also did not affect the interaction between RodA and PBP2 as their $Ef_A$ value in the presence of a third plasmid expressing MreD was 13.3 ± 2.9% (n = 9), which was not significant different from their $Ef_A$ of 11.8 ± 1.4% (n = 3) in the presence of the third, empty plasmid (S4D Fig). These combined results suggest a regulatory mechanism by which MreC interacts with PBP2 and changes its conformation, while MreD interacts with MreC and PBP2 to prevent this conformational change of PBP2. These conformational changes could correspond the proposed on and off states of PBP2 as published [39] (Fig 4A and 4B).

**Table 2. Summary of the calculated acceptor FRET (Ef$_A$) efficiencies from spectral FRET measurements for listed samples.**

| Parameter | FRET pairs | | Third plasmid | | | | | | | | |
| | | | Empty vector | | | MreC | | | MreCD | | |
| | pTHV037 | pSAV057 | Ef$_A$ (%) | SD (%) | N | Ef$_A$ (%) | SD (%) | N | Ef$_A$ (%) | SD (%) | N |
|---|---|---|---|---|---|---|---|---|---|---|---|
| **Positive control** | | | | | | | | | | | |
| Tandem | Empty vector | mKO-mCh | 30.4 | 1.9 | 4 | 31.3 | 3.2 | 3 | 31.7 | 4.3 | 2 |
| **Negative control** | | | | | | | | | | | |
| RodA+GlpT | mCh-RodA | mKO-(GGS)$_2$-GlpT | 2.4 | 1.3 | 7 | 2.2 | 1.1 | 2 | 1.4 | 2.1 | 4 |
| **Biological interactions** | | | | | | | | | | | |
| RodA+PBP2$^{WT}$ | mCh-RodA | mKO-PBP2$^{WT}$ | 8.8 | 1.1 | 6 | 4.9 | 0.6 | 2 | 9.2 | 1.5 | 4 |
| RodA+$^{MalFNT}$PBP2 | mCh-RodA | mKO-$^{MalFNT}$PBP2 | 7.6 | 1.8 | 5 | 4.2 | 0.3 | 2 | 5.5 | 1.7 | 4 |
| PBP2$^{WT}$+PBP2$^{WT}$ | mCh-PBP2$^{WT}$ | mKO-PBP2$^{WT}$ | 10.6 | 2.1 | 6 | 8.6 | 1.4 | 2 | 9.8 | 1.2 | 4 |
| PBP2$^{WT}$+$^{MalFNT}$PBP2 | mCh-PBP2$^{WT}$ | mKO-$^{MalFNT}$PBP2 | 9.4 | 3.5 | 5 | 9.8 | 1.9 | 2 | 8.7 | 1.8 | 4 |

**N**, number of samples measured. See for all individual values S1 Data.

*p* values for comparisons are shown in the figures.

## The cytoplasmic part of PBP2 is important for the interplay with the MreCD proteins

As shown the cytoplasmic NT part of PBP2 has an essential unknown function rather than being involved in the RodA-PBP2 interaction (Fig 3). We considered that the NT of PBP2

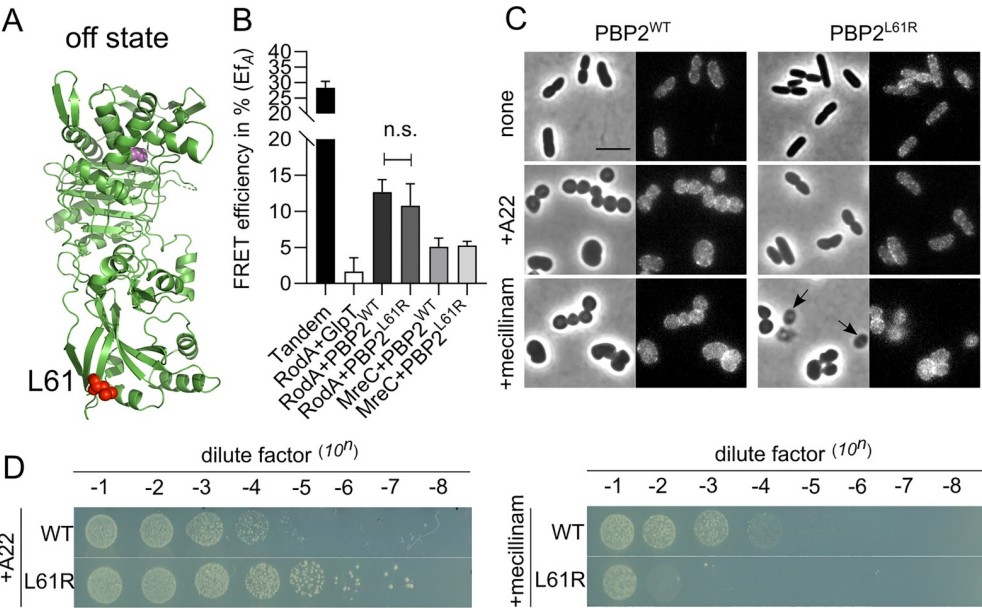

**Fig 5. Hyperactive mutant PBP2$^{L61R}$ is more sensitive to mecillinam. a.** Modeled "off state" structure of *E. coli* PBP2 from *H. pylori* PBP2 using Phyre 2. Structural information lacks on the juxta-membrane, transmembrane helix and cytoplasmic regions of PBP2. Residue Leu61 is colored in red and shown as spheres. Active site Ser330 is colored in pink and shown as spheres. **b.** Hyperactive mutant PBP2$^{L61R}$ interacts similar like PBP2$^{WT}$ with RodA and MreC. RodA and MreC were fused with mCherry, and PBP2$^{WT}$ and PBP2$^{L61R}$ were fused with mKO. **c.** Phase contrast and fluorescence images of cells expressing PBP2$^{L61R}$ were less sensitive to A22 but hypersensitive to mecillinam in liquid culture. LMC500 strain expressing nothing, or PBP2$^{WT}$, or PBP2$^{L61R}$ were grown in LB medium at 37°C. IPTG induction (15 μM), and A22 treatment (10 mg·L$^{-1}$) or mecillinam treatment (2 mg·L$^{-1}$) were applied to each culture for 2 mass doublings. Arrows indicate cells that lysed after mecillinam treatment in the PBP2$^{L61R}$ culture. Scale bar equals 5 μm. **d.** Spot assay to test the sensitivities of PBP2$^{WT}$ and PBP2$^{L61R}$ to A22 (10 mg·L$^{-1}$) and mecillinam (2 mg·L$^{-1}$).

might be important for its self-interaction and or interactions with other partner proteins. However, the $Ef_A$ values of $^{MalFNT}$PBP2 with wild-type PBP2, with MreC and with MreD were not different from those of wild-type PBP2 (Figs 4C and S7 and Table 1). Interestingly, the $Ef_A$ value of the interaction between MreC and wild-type PBP2, but not the $^{MalFNT}$PBP2, was reduced by the co-expression of MreD (Figs 4C and S4C and Table 1). Similarly, in the three-plasmid FRET experiments, MreD was not able to suppress the MreC-mediated change in the $^{MalFNT}$PBP2+RodA interaction; the $Ef_A$ value of RodA+$^{MalFNT}$PBP2 FRET remained at 5.5 ± 1.7% in presence of MreCD, rather than being restored to 9.2 ± 1.5% as in the RodA+-PBP2$^{WT}$ experiments (Figs 4D and S4C, Table 2). Together, these results indicate that the cyto-plasmic part of PBP2 plays a role in the MreCD-mediated regulation of the interaction between PBP2 and RodA.

## MreCD proteins do not alter PBP2 self-interaction

So far our results have shown that MreC and MreD have opposite effects on the interaction between RodA and PBP2. In contrast, the interaction between two PBP2 molecules [18] was not significantly affected upon overexpression of MreC or MreCD, as the calculated $Ef_A$ values for the PBP2+PBP2 interaction remained unchanged compared to the values for the expression of the third empty plasmid (Fig 4D and Table 2). RodA was also found to interact with itself (Table 1 and S7 Fig). Likely PBP2 and RodA function as a complex of dimers, which might also allow simultaneously synthesis of multiple glycan strand as has been proposed [51–53].

## Effect of mecillinam on the interaction between RodA and PBP2

As shown above, both the transmembrane helix and periplasmic part of PBP2 contribute to its interaction with RodA (Fig 3). The binding of MreC to the periplasmic hydrophobic zipper domain of PBP2, which presumably changes the conformation of PBP2 from off state to on state, reduces the detected $Ef_A$ between RodA and PBP2 (Fig 4D and 4E). Interestingly, the PBP2 specific inhibitor mecillinam, which binds the TPase active site of PBP2, also caused a reduction in the FRET efficiency of the interaction between RodA and PBP2 pairs including the domain swap variants and the PBP2 inactive variant (see below). The RodA+PBP2$^{WT}$ interaction pair yielded a reduced $Ef_A$ of 7.9 ± 1.6% in the presence of mecillinam [34], com-paring with the $Ef_A$ of 12.7 ± 1.7% without mecillinam (Figs 2 and S8, Table 1). Similarly, the interaction pair of RodA+$^{MalFNT}$PBP2 yielded a reduced $Ef_A$ of 6.2 ± 1.1% (Table 1 and S8 Fig). A possible explanation for the reduced but not abolished interaction between RodA and PBP2 by mecillinam could be that mecillinam blocks PBP2 in the active conformation, which changes the interaction in the periplasmic parts of the proteins while the transmembrane regions of RodA and PBP2 remain interacting (Fig 3). Replacing the transmembrane helix of PBP2 to abolish this part of the interaction (RodA+$^{MalF37}$PBP2), mecillinam further reduced the $Ef_A$ value to 3.0 ± 2.0% (S8 Fig and Table 1), consistent with an almost complete loss of the interaction between RodA and $^{MalF37}$PBP2. That mecillinam causes disruption of only one out of two interacting regions between PBP2 and RodA is also in agreement with the observations that it does not disrupt the structure of the elongasome [54]. The inactive mutant PBP2$^{S330C}$, which cannot bind benzylpenicillin covalently [37], still responded to mecillinam and showed a similar $Ef_A$ reduction as PBP2$^{WT}$ (S8 Fig and Table 1), suggesting that the mutant can still interact with the antibiotic at its active site. Alternatively, mecillinam binds to a second (allo-steric) site that activates the protein as has been reported for some β-lactams and PBPs [55–57]. High concentrations of aztreonam that do not inhibit PBP2 cause an increase in binding of the fluorescent β-lactam Bocillin FL [58], suggesting that an allosteric site might be present in PBP2. Two recently published crystal structures of *E. coli* PBP2 in the presence of

avibactams show only binding to the active site [59] indicating that at least these antibiotics do not bind to an allosteric site at the used concentrations. The change in FRET efficiency upon binding of mecillinam, which is measured in the cytoplasm, suggests conformational changes in the complex that are transmitted via the transmembrane areas. Based on the crystal structures of PBPs with allosteric sites, such a site is expected to be at the interface between V-shaped MreC-binding domain of PBP2 and the globular β-lactam binding domain [55, 56]. An allosteric site would make sense for a TPase that has to bind two peptide side chains to be able to make the crosslink. Allosteric activation of the TPase reaction would make sure that the protein is only active in the presence of a disaccharide peptide that is in the process of being transglycosylated by RodA. Our observation that mecillinam binding caused a reduction in FRET efficiency similar to the presence of MreC, even of the active site mutant S330C, might plead for the presence of an allosteric site in PBP2.

## PBP2$^{L61R}$ activates RodA without changing the interaction of RodA and PBP as measured by FRET

A recent study reported a version of PBP2 in which Leu61 was replaced by Arg (PBP2$^{L61R}$) that could suppress an MreC defect, and was proposed to stay in the on state conformation mimicking activation by MreC [40]. If this were the case, the RodA+PBP2$^{L61R}$ pair would be expected to have a reduced FRET efficiency, since the MreC activated RodA+PBP2$^{WT}$ pair resulted in a reduction in FRET efficiency (Fig 4D). Therefore, we constructed an N-terminal mKO fusion of PBP2$^{L61R}$ to test the interactions with its partner proteins. Surprisingly, the Ef$_A$ for RodA+PBP2$^{L61R}$ remained 10.8 ± 3.0%, which was not significantly different from the Ef$_A$ of RodA+PBP2$^{WT}$ that was presumably in the off state (Figs 5A, 5B and S9A, Table 1). Mecillinam reduces the Ef$_A$ value of the RodA+PBP2$^{L61R}$ interaction to 6.3 ± 3.5%, which was also comparable with its effect on the RodA+PBP2$^{WT}$ interaction (Table 1 and S8 Fig). Unfortunately, the co-expression of PBP2$^{L61R}$ together with either MreC or MreD alone, or both together, was not possible in most cases, as FRET cells repeatedly lost the mKO-PBP2$^{L61R}$ signal upon induction (Table 1), suggesting toxicity of these combinations. The cells did not lose the mKO signal in only two out of six attempts to co-express MreC and PBP2$^{L61R}$. Of those samples the calculated Ef$_A$ value of the MreC and PBP2$^{L61R}$ pair remained at 5.4 ± 1.7%, which was comparable with MreC+PBP2$^{WT}$ (Fig 4A and Table 1). These results suggest that the hyperactive mutant PBP2$^{L61R}$ likely behaves similarly as wild-type PBP2 in the interaction with its partner proteins.

Having observed these unexpected results, we continued to further characterize the hyperactive PBP2$^{L61R}$. Consistent with its reported functionality [40], we observed that mKO-PBP2$^{L61R}$ was capable to support growth of the PBP2(TS) strain LMC582 at the non-permissive temperature (S9B Fig). However, the expression of mKO-PBP2$^{L61R}$ resulted in longer and thinner cells (Figs 5C and S9C and S3 Data), and in reduced sensitivity of cells to the MreB inhibitor A22 (Fig 5C) as reported [40]. Interestingly, cells expressing PBP2$^{L61R}$ were hypersensitive to mecillinam (Fig 5C and 5D). These results indicate potential defects in the peptidoglycan layer of cells expressing PBP2$^{L61R}$, and these defects may be tolerable under undisturbed growth conditions but are exacerbated in the presences of mecillinam. *In vitro* peptidoglycan synthesis experiments showed that unbalancing the TPase and GTase activities of PBP1A-PBP2 complex by inactivation of PBP2 with mecillinam resulted in longer glycan chains [19], hence we wondered whether the presence of PBP2$^{L61R}$ affected peptidoglycan synthesis in the cell. We prepared peptidoglycan and analyzed its composition from cells expressing PBP2$^{L61R}$, wild-type PBP2, and the control membrane protein GlpT. As predicted, the peptidoglycan from all strains retained a similar extent of peptide cross-linkage. By contrast, only the peptidoglycan from the PBP2$^{L61R}$ expressing cells contained unusually long glycan chains with

**Table 3. Summary of muropeptide composition of LMC500 strains carrying no plasmid or different expression plasmids.**

| Muropeptides[2] or feature | Percent peak area (%)[1] | | | | |
|---|---|---|---|---|---|
| | LMC500 | LMC500 mKO | LMC500 mKO-PBP2[WT] | LMC500 mKO-PBP2[L61R] | LMC500 mKO-GlpT |
| Monomers (total) | 52.6 ± 0.1 | 52.0 ± 0.0 | 52.1 ± 1.1 | 52.2 ± 0.9 | 52.2 ± 0.0 |
| dipeptides | 1.4 ± 0.0 | 1.6 ± 0.1 | 1.3 ± 0.1 | 1.2 ± 0.0 | 1.4 ± 0.0 |
| tripeptides | 5.0 ± 0.2 | 5.0 ± 0.0 | 5.0 ± 0.5 | 4.0 ± 0.3 | 4.7 ± 0.1 |
| tetrapeptides | 41.6 ± 0.0 | 41.0 ± 0.0 | 41.3 ± 0.5 | 43.3 ± 2.0 | 41.8 ± 0.2 |
| anhydro | 0.9 ± 0.0 | 1.0 ± 0.0 | 1.0 ± 0.0 | 0.7 ± 0.0 | 1.0 ± 0.0 |
| LysArg | 3.7 ± 0.4 | 3.4 ± 0.2 | 3.5 ± 0.5 | 3.0 ± 1.2 | 3.4 ± 0.5 |
| Dimers (total) | 40.5 ± 0.0 | 40.8 ± 0.0 | 40.2 ± 0.7 | 41.7 ± 0.9 | 40.7 ± 0.1 |
| tetratripeptide | 3.1 ± 0.3 | 3.0 ± 0.2 | 3.0 ± 0.1 | 2.2 ± 0.0 | 2.9 ± 0.1 |
| tetratetrapeptide | 36.0 ± 1.0 | 36.3 ± 0.2 | 35.0 ± 2.6 | 37.1 ± 2.7 | 36.0 ± 0.9 |
| anhydro | 2.0 ± 0.0 | 2.1 ± 0.0 | 2.2 ± 0.0 | 1.7 ± 0.0 | 2.0 ± 0.0 |
| LysArg | 1.6 ± 0.0 | 1.7 ± 0.1 | 1.5 ± 0.2 | 1.2 ± 0.2 | 1.6 ± 0.1 |
| Trimers (Total) | 4.4 ± 0.1 | 4.6 ± 0.1 | 4.8 ± 0.0 | 3.8 ± 0.0 | 4.5 ± 0.1 |
| Dipeptides (total) | 1.4 ± 0.0 | 1.6 ± 0.1 | 1.3 ± 0.1 | 1.2 ± 0.0 | 1.4 ± 0.0 |
| Tripeptides (total) | 7.2 ± 0.9 | 7.2 ± 0.2 | 7.6 ± 1.1 | 6.3 ± 0.7 | 7.0 ± 0.3 |
| Tetrapeptides (total) | 85.8 ± 0.1 | 85.8 ± 0.0 | 85.3 ± 0.1 | 87.7 ± 0.0 | 86.2 ± 0.0 |
| Chain ends (anhydro) | 2.3 ± 0.0 | 2.4 ± 0.0 | 2.5 ± 0.0 | 1.9 ± 0.0 | 2.3 ± 0.0 |
| Glycan chain length (DS)[3] | 42.8 ± 0.2 | 40.5 ± 0.0 | 38.3 ± 0.1 | **52.1 ± 0.5** | 41.6 ± 0.0 |
| Degree of cross-linkage | 23.2 ± 0.1 | 23.4 ± 0.0 | 23.3 ± 0.1 | 23.3 ± 0.2 | 23.3 ± 0.0 |
| % Peptides in cross-links | 47.4 ± 0.1 | 48.0 ± 0.0 | 48.0 ± 1.1 | 47.8 ± 0.9 | 47.8 ± 0.0 |

[1] values are mean ± variation of two biological replicates.

[2] muropeptide names according to Glauner, 1988 [60].

[3] average glycan chain length in disaccharide (DS) units calculated from the percentage of anhydro-MurNAc containing muropeptides. The bold number highlights the increased average glycan chain length in cells expressing PBP2[L61R].

a mean length ~52 disaccharide units (Table 3). The peptidoglycan of the strain overexpressing wild-type PBP2 had a mean glycan chain length of ~38 units, and the mean glycan chain lengths of the other strains were between 40–43 disaccharide units (Table 3, S1 Table). Together with the evidence that PBP2[L61R] stimulates the GTase activity of RodA in vitro [40] and our results on the cellular interactions, it was likely that the L61R exchange in PBP2 enhances only the activity of RodA and has no effect on PBP2's TPase activity. A stimulating effect of PBP2[L61R] on RodA's GTase activity would be consistent with the previously observed A234T mutation in RodA that suppressed the morphological defects of MreC mutants [40], and would also explain why PBP2[L61R] could only poorly restore survival and rod-shape in cells depleted of MreCD or RodZ [40]. The tolerance to A22 and changes in MreB dynamics in the PBP2[L61R] background could also be explained by the enhanced RodA GTase activity by PBP2[L61R], since a direct interaction between RodA and MreB was detected with a $Ef_A$ value of 7.2 ± 3.4% (Table 1 and S10 Fig). All together the changes in cell morphology, the resistance to A22 and sensitivity to mecillinam, the partial compensation of MreCD-RodZ depletion and MreB dynamic changes are all likely due to an enhanced GTase activity of RodA (and perhaps PBP1A) [19, 40], which results into longer glycan chains in the peptidoglycan mesh.

## Discussion

In this work we aimed to reveal the regulation of peptidoglycan synthesis during length growth. We showed that RodA and PBP2 form a stable subcomplex independent of their

biochemical activities (Fig 2). This interaction requires the transmembrane helix and periplasmic parts of PBP2 (Fig 3), which is confirmed by the RodA-PBP2 crystal structure [48]. Replacing the transmembrane region of PBP2, largely disturbed their interaction whereas the absence of the cytoplasmic domain had no effect (Fig 3).

Our FRET experiments reveal that MreC interacts directly with PBP2, which is in agreement with the published structural [39] data and also detected for first time the interaction between MreD and PBP2 (Fig 4). MreD showed a negative regulation on the interaction between MreC and PBP2 interaction, as their interaction was lost in presence of MreD (Fig 4). The cytoplasmic domain of PBP2 is needed for the interaction with MreD. When this domain is replaced by that of MalF, MreD can no longer prevent the Interaction of MreC with PBP (Fig 4). Interestingly, our *in vivo* FRET experiments revealed that MreC modulates the interaction between PBP2 and RodA. The presence of MreC reduced the RodA-PBP2 FRET efficiency (Fig 4). Expression of MreC and MreD restored the RodA-PBP2 FRET efficiency to the level in the absence of MreC, whereas MreD on its own did not affect the interaction between RodA and PBP2. This suggest that MreC caused a conformational change in the RodA-PBP2 complex that could be prevented by the presence of MreD. Structural data show that the interaction between MreC and PBP2 causes a conformational change in PBP2 that was suggested to correspond to its activation from the off state to the on state [39]. Combining our results and published evidences, this conformational change could correspond to the change in the interaction between PBP2 and RodA induced by MreC. And likely, when MreD is co-expressed with MreC, the reversed change in the interaction between PBP2 and RodA could correspond to the PBP2 conformational change from the "on" state to the "off" state. This potential of MreC and MreD to affect the RodA-PBP2 interaction could reflect part of the regulation of elongasome activity and peptidoglycan synthesis during length growth.

The presence of mecillinam gave also a reduction in the FRET efficiency of the RodA PBP2 complex similar to the presence of MreC. Even the active site variant S330C of PBP2 in combination with RodA gave a comparable reduction in FRET efficiency. Indicating that the conformation change induced by meciliinam must propagate through the transmembrane helices of the complex. The binding of mecillinam to an allosteric mecillinam binding site close to the V-shaped domain that interacts with MreC [55, 56] would perhaps make more sense than the propagation of a conformational change in the protein due to binding of mecillinam to the classical penicillin binding globular domain. Some evidence that PBP2 can be allosterically activated comes from the work of Kocaoglu et al [58] where it was shown that high concentrations of aztreonam increases the amount of fluorescent Bocillin binding by PBP2. Although MreC as well as mecillinam causes a reduction in the FRET efficiency between RodA and PBP2, the accompanying conformational changes could be different.

The PBP[L61R] mutants was published to partially complement defects of the absence of MreC, and proposed to be in a different structural conformation from wild-type [40]. Therefore it was expected that this mutant in combination with RodA would have a similar FRET efficiency as PBP2-RodA in complex with MreC. However, PBP2[L61R] behaved exactly as the wild type with respect to its interaction with RodA and also in the presence of mecillinam. Therefore, the enhanced GTase activity of RodA by this mutant [40] as illustrated by the increase in glycan strand length and the longer and thinner cell morphology is likely not equivalent to the impact MreC has on the interaction between RodA and PBP2. The PBP2[L61R] could also complement the MreD and RodZ deletion [40], indicating that it is able to operate without being regulated by MreCD. Because a combination of PBP2[L61R], RodA and MreC expressed from plasmid appeared to be very toxic, we assume that the effect of MreC is additive causing uncontrolled activity of the Rod complex.

Interestingly, this regulation of MreCD proteins on PBP2 activity is formally similar to the recently proposed model for the regulation of septal peptidoglycan synthesis [22]. In that model, the FtsBLQ subcomplex inhibits the activities of PBP3 (consequently also inhibiting FtsW) and PBP1B, and keeps septal peptidoglycan synthesis in check. A small amount of FtsN is already present at pre-septal sites together with ZipA and the class A PBPs, PBP1A and PBP1B [61]. However, only once FtsN accumulates at higher levels it is able to relieve the suppression of FtsBLQ on the peptidoglycan synthases, thereby activating septal peptidoglycan synthesis. As mentioned in the introduction, elongasome and divisome share many similarities in components and function. When comparing the cellular numbers of these proteins synthesized per generation [50], we also noticed that the average number of FtsN molecules per cell is about 2 times higher than that of FtsBLQ and FtsW-PBP3 proteins in different growth media. The number of MreC molecules is also about 2 times higher than that of MreD and PBP2-RodA molecules [50]. These numbers are based on ribosome profiling and an estimated average protein lifetime, which could be different for specific proteins in the cells. However, they do suggest a difference in the mentioned protein rations and our data suggest that balanced expression of MreC and MreD is important. Moreover, MreC, like FtsN, is also reported to interact with itself, and possibly accumulates at the elongasome.

Based on our observations and these published data, we propose a model for the possible regulation mechanism of PBP2 (elongasome) activity and cylindrical peptidoglycan synthesis (Fig 6). The peptidoglycan synthases RodA and PBP2 form a stable subcomplex. MreC stimulates and activates PBP2 and RodA, while MreD interferes with the PBP2-MreC interaction to keep PBP2 activity in check (Fig 6, left). Likely, RodA could have some basic constitutive GTase activity, since RodA alone, or together with inactive PBP2, showed lipid II polymerization activity [10, 40]. The further binding and accumulation of MreC to the elongasome would eventually outcompete MreD, which will activate PBP2 and consequently balance the polymerization and crosslinking actions, and initiate robust peptidoglycan synthesis (Fig 6, right). Because hydrolytic activity is required to allow insertion of newly synthesized peptidoglycan

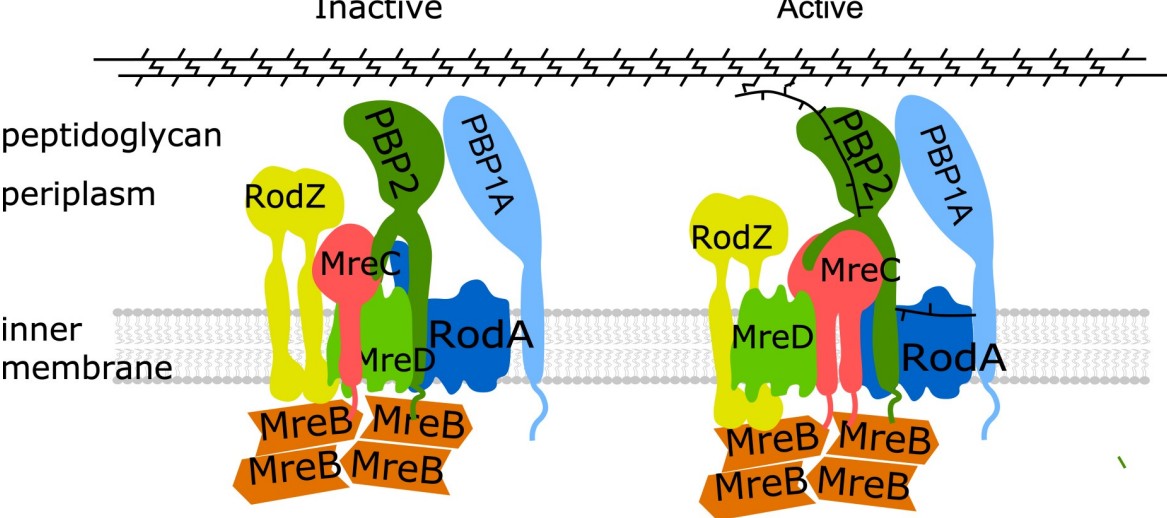

**Fig 6. Model for regulation of elongasome and PG synthesis.** RodA and PBP2 interact with each other and form a stable subcomplex. MreC and RodZ interact strongly with MreB filaments that likely link MreB to the PG synthesis proteins. MreC interacts with PBP2 that could stimulate and activate PBP2, while MreD, which interacts with both PBP2 and MreC, suppresses the activation of PBP2 by MreC, and keeps PG synthesis under control. The accumulation of MreC to the elongasome will finally abolish the inhibition of MreD and activate PBP2 by changing its conformation from the "off state" to the "on state", and subsequently activate and balance the PG synthesis of the elongasome.

into the existing mesh [62, 63], a balanced regulation is likely needed to avoid premature glycan strand synthesis. Consequently, the elonsasome, like the divisome, must have a mechanism to sense whether all partners are at the right position to act. A well-regulated moment for peptidoglycan synthesis by balancing the MreCD ration is likely part of such a regulatory system.

## Materials and methods

### Media, strains, plasmids and primers

LB (10 g tryptone, 5 g yeast extract and 10 g NaCl, per liter) and Gb4 (6.33g $K_2HPO_4\cdot3H_2O$, 2.95g $KH_2PO_4$, 1.05 g $(NH_4)_2SO_4$, 0.10 g $MgSO_4\cdot7H_2O$, 0.28 mg $FeSO_4\cdot7H_2O$, 7.1 mg Ca $(NO_3)_2\cdot4H_2O$, 4 mg thiamine, 2 mg uracil, 2 mg lysine, 2 mg thymine, and 0.5% glucose, per liter, pH 7.0) were used for cell cultures in rich and minimal medium, respectively, as indicated. Final concentrations for antibiotics were: 100 $\mu g\cdot L^{-1}$ ampicillin, 50 $\mu g\cdot L^{-1}$ kanamycin and 25 $\mu g\cdot L^{-1}$ chloramphenicol.

*E. coli* strains and plasmids used in this study are listed in S2 Table. Primers used in this study were listed in S3 Table. The plasmids were constructed as following:

**pXL29.** Plasmid pXL28 and pWA004 were digested with *EcoR*I and *Hind*III restriction enzymes, the generated *(*GGS)$_2$-GlpT expressing gene and pSAV057-mKO linear vector were ligated together to generate the mKO-(GGS)$_2$-GlpT expressing plasmid.

**pXL36, pXL40, pXL44, pXL48, pXL56 and pXL63.** Plasmids pXL36 and pXL40 expressing mCherry-fused RodA$^{R109A}$ and RodA$^{Q207R}$ were generated from pSAV047-RodA by mutagenesis PCR using primer pairs priXL61-priXL61 and priXL69-priXL70, respectively. To construct non-fused version of RodA variants, wild-type *rodA* gene was amplified using primer priXL59 and priXL60 from the MG1655 genomic DNA and ligated into empty pSAV057 vector, to generate plasmid pXL63. The two mutants plasmids were generated in the same way as described for pXL63. mKO fused RodA plasmid pXL56 was constructed by cutting and pasting the *rodA* gene from pSAV047-RodA to the pSAV058 plasmid with *EcoR*I and *HindIII* restriction enzymes.

**pXL148, pXL149, pXL158 and pXL159.** The PBP2 domain swap plasmids were constructed by Gibson assembly [64]. For N-terminus replacement, primer pairs priXL146-priXL258 and priXL147-priXL259 were used for PCR from plasmid pWA004. The PCR products were purified after *DpnI* digestion and assembled to generated pXL148 that excludes the N-terminus of PBP2. For pXL149, the primer pair priXL258-priXL260 was used to amplify the entire pWA004 plasmid excluding the first 45 residues. Primer pair priXL261 and priXL263 was use to amplify the first 37 residues of MalF from MG1655 genome. The PCR products were purified after *DpnI* digestion and assembled to generate pXL149 plasmid. The PBP2$^{S330C}$ and PBP2$^{L61R}$ plasmids were constructed with mutagenesis PCR from the pWA004 plasmid using primer pairs priXL274-priXL275 and priXL276-priXL277, respectively.

**pXL165, pXL166 and pXL169.** *mreC*, *mreCD* and *mreD* genes were amplified from MG1655 genome using primer pairs priXL282-priXL286, priXL282-priXL283 and priXL299-priXL283, respectively, and cloned into plasmid pSAV047 with *EcoR*I and *HindIII* restriction enzymes, to generate the mCherry fused version of these genes.

**pXL167 and pXL168.** The third plasmids used in the three-plasmids FRET experiments. *mreC* and *mreCD* genes were cloned into plasmid pSG4K5 [65] with Gibson assembly, respectively, and the p$_{trcdown}$ promoter was introduced to control the protein expression. Primer pair priXL294-priXL295 was used to amplify the linear vector from pSG4K5. Primer pair priXL296-pp15 was used to amplify the p$_{trcdown}$ promoter. Primer pairs pXL284-priXL297 and priXL284-priXL298 were used to amplify the *mreCD* and *mreD* genes, respectively.

## Bacterial growth, morphology and protein localization

For general growth experiments in rich medium, overnight cultures (37˚C) were diluted 1:1000 into fresh LB medium with 0.5% glucose and the required antibiotics, and grew to $OD_{600}$ around 0.2 at 37˚C. Cultures were further diluted 1:5 into fresh LB medium with required antibiotics, and induced with 15 μM IPTG for 2 mass doubling at 37˚C ($OD_{600}$ reached around 0.2).

For complementation experiments, temperature sensitive strains expressing the mutant plasmids were grown as described above at 30˚C. Cultures were further diluted 1:5 into fresh LB medium with required antibiotics, and induced with 15 μM IPTG for 2 mass doubling at 30˚C and 42˚C, respectively ($OD_{600}$ reached around 0.2).

After induction, cells were fixed with FAGA (2.8% formaldehyde and 0.04% glutaraldehyde, final concentration) for 15 minutes and centrifuged at 7000 rpm for 10 min at room temperature. Cell pellets were suspended and washed 3 times with PBS (pH 7.2) buffer. Subsequently, bacterial morphology and protein localization were imaged by wide field phase contrast and fluorescence microscopy. Specially, cells expressing the mKO fused proteins were firstly matured at 37˚C overnight before imaging by microscopy.

## FRET experiment and data analysis

Protein interactions were detected by FRET as described previously [34, 41, 49, 66]. For the FRET experiments, mCherry and mKO fluorescent proteins were used as acceptor and donor fluorophores, respectively. LMC500 strain was co-transformed with the FRET pairs that were to be detected. In each FRET experiment, the empty-vector reference, mCherry reference, mKO reference were included to be able to calculate the $Ef_A$ by unmixing of the measured FRET pair spectrum in its individual components; background, mCherry, mKO and sensitized emission spectra. A tandem fusion of mKO-mCherry was used as positive control, and the mCherry-RodA and mKO-GlpT pair was used as negative control. After transformation, FRET strains were firstly grown in LB medium (with antibiotics and 0.5% glucose) overnight at 37˚C, and diluted 1:1000 into fresh medium and grown to $OD_{600}$ around 0.2 at 37˚C. Subsequently, FRET strains were diluted 1:500 into Gb4 medium and grown to steady state at 28˚C ($OD_{450}$ was kept below 0.2). All FRET strains were induced with 15 μM IPTG (and treated with mecillinam at 2 mg·L$^{-1}$ concentration as indicated) for two mass doubling before FAGA fixation. After fixation, FRET cells were pelleted by centrifugation at 7000 rpm at room temperature and washed 3 times with PBS buffer (pH 7.2). Then all samples were incubated at 37˚C overnight and stored at 4˚C for 1 extra day before measured with spectrofluorimeter (Photon Technology International, NJ). Emission spectra of acceptor and donor fluorophores were measured through 6-nm slit widths with 1 second integration time per scanned nm for 3 times averaging. Filters 587/11 nm (587/11 nm BrightLine single band-pass filter, Semrock, New York, NY, USA) and 600nm long-pass (LP) filter (Chroma Technology Corp., Bellows Falls, VT) were used for excitation and emission of acceptor fluorophore (mCherry), while 541/12 nm (Semrock) and 550 nm long pass (Chroma) filters were used for mKO excitation and emission, respectively. For calculation, measurement of PBS buffer was subtracted from all samples, and the empty-cell reference was subtracted from the donor and acceptor spectra. The FRET efficiencies were calculated as described previously [41, 49, 66].

For three plasmids FRET, a third plasmid (expressing MreC, MreD or expressing MreCD both) was introduced into the whole two plasmids FRET system. Empty pSG4K5 vector was also introduced as a control to correct for the reduction in FRET efficiency due to the burden of maintaining three plasmids.

## Spot assay

To test the sensitivity of *E. coli* strains to A22 and mecillinam, LMC500 strain was transformed with pWA004 (PBP2$^{WT}$) or pXL159 (PBP2$^{L61R}$). Strains expressing each construct were grown in LB medium as descried above without induction. Cell cultures were diluted with varying dilution factors (Fig 4C). A drop of 10 μl cell culture from each dilution was loaded on the LB agar dish (with chloramphenicol, 15 μM IPTG, 10 μg·mL$^{-1}$ A22 or 2 μg·mL$^{-1}$ mecillinam) and incubated overnight at 37˚C.

## Peptidoglycan analysis

Peptidoglycan sacculi were prepared from *E. coli* cells, digested with cellosyl (kind gift from Hoechst, Germany), reduced with sodium borohydride and analyzed by high pressure liquid chromatography as described (Separation and quantification of muropeptides with high-performance liquid chromatography) [60].

## Microscopy

Bacterial cell samples were immobilized on 1.3% agarose pads (w/v in Gb4 medium) and imaged by microscopy. Fluorescence microscopy was carried out either with an Olympus BX-60 fluorescence microscope equipped with an UPlanApo 100×/*N.A.* 1.35 oil Iris Ph3 objective, or with a Nikon Eclipse Ti microscope equipped with a C11440-22CU Hamamatsu ORCA camera, a CFI Plan Apochromat DM 100× oil objective, an Intensilight HG 130W lamp and the NIS elements software (version 4.20.01). Images were acquired using the Micro Manager 1.4 plugin for ImageJ, and analyzed with Coli-Inspector supported by the ObjectJ plugin for ImageJ (version 1.49v) [67].

## Supporting information

**S1 Fig. Alignment of the amino acid sequences of RodA and FtsW from *E. coli*.** The two inactive RodA mutants were constructed based on the previous functional studies of FtsW [1, 2] (bold red and labeled with *). The protein sequence alignment was generated with the online tool Multiple Sequence Alignment (MUSCLE). Shadow colors indicate the average BLOSUM62 score of the paired residues: light blue> = 3 (identical amino acids), light gray> = 2 (similar amino acids), no color (different amino acids).
(TIF)

**S2 Fig. Overview of the spectral unmixing of all FRET data of interactions between RodA and PBP2 variants.** A. Inactive RodA and PBP2 variants shown in Fig 2B. B. Domain swap variants of PBP2 shown in Fig 3C. LMC500 cells expressing each FRET pair were grown in Gb4 medium to steady state at 28˚C and further cultured in the presence of 15 μM IPTG for 2 mass doublings. FRET pairs are listed above the spectra. For each pair the upper panel contains the measured spectrum excited at 538 nm in black dots, the calculated spectrum (in red) and its unmixed components; blue is background, magenta is mKO, orange is mCherry and green is the sensitized emission. The middle panel is the measured spectrum of mCherry excited at 590 nm in black dots, the calculated spectrum (in red) and its unmixed components; blue is background and magenta is mCherry. The bottom panel shows the residuals of the measured and calculated spectrum.
(TIF)

**S3 Fig. Overview of the spectral unmixing data of FRET measurements of the interactions between MreCD and PBP2 variants that are listed in Fig 4B.** LMC500 strain expressing each

FRET pair was grown in Gb4 medium to steady state at 28˚C and induced with 15 μM IPTG for 2 mass doublings. FRET pairs are listed above the spectra. MreC$^D$, MreC and MreD were expressed from one plasmid, and MreC was fused with mCherry while MreD was not fused. Panels as in S2 Fig.
(TIF)

**S4 Fig. Overview of the unmixing of the spectra of the FRET data of the three plasmids interaction groups that are listed in Fig 4D LMC500 cells expressing each FRET pair were grown in Gb4 medium to steady state at 28˚C and further cultured in the presence of 15 μM IPTG for 2 mass doublings.** FRET pairs are listed above the spectra. A, B, C and D. Unmixing data of FRET experiments in the presence of the third: empty vector (EV), or MreC (expressing MreC alone from the third plasmid under a P$_{trcdown}$ promoter), MreCD (expressing MreCD together from the third plasmid under a P$_{trcdown}$ promoter), or MreD alone from the third plasmid, respectively. Panels as in S2 Fig.
(TIF)

**S5 Fig. Co-expression of MreD suppresses the morphological defects of MreC overexpression.** A. Phase contrast (upper panels) and fluorescent (bottom panels) images of the morphology and FP-protein localization, respectively, of LMC500 cells expressing mCherry (control), mCherry-MreC, mCherry-MreD or mCherry-MreCD. MreCD were expressed from one plasmid and MreC was fused with mCherry, while MreD was non-fused. Cells were growth in LB medium at 37˚C. B and C. Images of the morphology of LMC500 cells with empty vector, expressing MreC, MreD or MreCD grown in LB at 37˚C or in minimal glucose medium (GB4) at 28˚C. Expression was induced for two mass doublings with 15 μM IPTG. Scale bar equals 5 μm.
(TIF)

**S6 Fig. MreD topology predictions generated from different prediction programs as indicated.** Numbers: MreD amino acid position. i: in the cytoplasm. o: in the periplasm. M: transmembrane sequences.
(TIF)

**S7 Fig. Self-interaction of PBP2 and RodA detected by FRET.** LMC500 cells expressing each FRET pair were grown in Gb4 medium to steady state at 28˚C and further cultured in the presence of 15 μM IPTG for 2 mass doublings. A. Acceptor FRET efficiency (Ef$_A$) between PBP2$^{WT}$ and PBP2$^{WT}$ calculated from the spectral FRET measurements. B. Overview of the spectral unmixing data of all the FRET pairs of self-interacting PBP2 and its variants. C and D. Calculated acceptor FRET efficiency (Ef$_A$) and spectral unmixing data of RodA self-interaction. FRET pairs are listed above the spectra. Panels as in S2 Fig. P value determined with Student's t-test (n.s.: not significant; (*: $p < 0.05$)).
(TIF)

**S8 Fig. Interaction between RodA and PBP2 variants in the presence of mecillinam.** LMC500 cells expressing each FRET pair were grown in Gb4 medium to steady state at 28˚C and further cultured in the presence of 15 μM IPTG and treated with mecillinam (2 μg·mL$^{-1}$) for 2 mass doublings. a. Acceptor FRET efficiency (Ef$_A$) calculated from the spectral FRET measurements. P value determined with Student's t-test (**: $p < 0.01$). b. Overview of the unmixing data of all the FRET samples showed in S8A Fig. FRET pairs are listed above the spectra. Panels as in S2 Fig.
(TIF)

**S9 Fig. Hyperactive PBP2L61R changes cell morphology without influencing its interaction with MreC or RodA.** A. Overview of the spectral unmixing data of all the FRET groups between hyperactive PBP2$^{L61R}$ and its interaction partners that are listed in Fig 5B. LMC500 cells expressing each FRET pair were grown in Gb4 medium to steady state at 28˚C and further cultured in the presence of 15 μM IPTG for 2 mass doublings. FRET pairs are listed above the graphs. Panels as in S2 Fig. B. Phase contrast and fluorescence images of the complementation of cells harboring a PBP2 temperature sensitive allele with PBP2$^{WT}$ or hyperactive PBP2$^{L61R}$. PBP2 temperature sensitive strain LMC582 was transformed with empty vector (EV), mKO-PBP2$^{WT}$ plasmid, or mKO-PBP2$^{L61R}$ plasmid and grown in LB medium at 30˚C (left panels) and 42˚C (right panels), respectively, and the expression of PBP2 versions was induced with 15 μM IPTG for 2 mass doublings. Scale bar equals 5 μm. C. Expression of the hyperactive PBP2$^{L61R}$ results into (slightly) longer and thinner cells. LMC500 strain was transformed with plasmid expressing either mKO-PBP2$^{WT}$ or mKO-PBP2$^{L61R}$, grown in LB at 37˚C and further cultured in the presence of 15 μM IPTG for 2 mass doublings. Cells were fixed and imaged by microscopy. Over 1000 cells were measured for statistical analysis.
(TIF)

**S10 Fig. Direct interaction between RodA and MreB detected with FRET.** LMC500 cells expressing each FRET pair were grown in Gb4 medium to steady state at 28˚C and further cultured in the presence of 15 μM IPTG for 2 mass doublings. A. Acceptor FRET efficiency (Ef$_A$) calculated from the spectral FRET measurements. P value determined with Student's t-test (**: p<0.01). B. Overview of the unmixing data of all the FRET samples showed in S10A Fig. FRET pairs are listed above the spectra. Panels as in S2 Fig.
(TIF)

**S1 Table. Muropeptide composition of LMC500 strains carrying no plasmid or different expression plasmids.**
(DOCX)

**S2 Table. Strains and plasmids used in this study.**
(DOCX)

**S3 Table. Primers used in this study.**
(DOCX)

**S1 Data. All FRET data Liu et al.**
(XLSX)

**S2 Data. Morphological parameters of MreCD expressing cells.**
(XLSX)

**S3 Data. Morphological paremeters of L61R expressing cells.**
(XLSX)

## Acknowledgments

We thank Jolanda Verheul for technical assistance, Lisa Atkinson for preparation of peptidoglycan sacculi and Patricia Dawn Adela Rohs and prof Thomas Bernhardt for the gift of modeled *E. coli* PBP2 structures.

## Author Contributions

**Conceptualization:** Xiaolong Liu, Tanneke den Blaauwen.

**Data curation:** Xiaolong Liu, Tanneke den Blaauwen.

**Formal analysis:** Xiaolong Liu, Jacob Biboy, Waldemar Vollmer, Tanneke den Blaauwen.

**Funding acquisition:** Xiaolong Liu, Waldemar Vollmer, Tanneke den Blaauwen.

**Investigation:** Xiaolong Liu, Jacob Biboy, Elisa Consoli.

**Methodology:** Xiaolong Liu, Waldemar Vollmer, Tanneke den Blaauwen.

**Project administration:** Tanneke den Blaauwen.

**Resources:** Xiaolong Liu, Tanneke den Blaauwen.

**Software:** Xiaolong Liu, Tanneke den Blaauwen.

**Supervision:** Waldemar Vollmer, Tanneke den Blaauwen.

**Validation:** Xiaolong Liu, Tanneke den Blaauwen.

**Visualization:** Xiaolong Liu, Tanneke den Blaauwen.

**Writing – original draft:** Xiaolong Liu, Tanneke den Blaauwen.

**Writing – review & editing:** Xiaolong Liu, Jacob Biboy, Elisa Consoli, Waldemar Vollmer, Tanneke den Blaauwen.

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
