## [Decision Letter · Decision Letter 0]

6 Jan 2020

Dear Dr den Blaauwen,

Thank you very much for submitting your Research Article entitled 'MreC and MreD balance the interaction between the elongasome proteins PBP2 and RodA' to PLOS Genetics. Your manuscript was fully evaluated at the editorial level and by independent peer reviewers. The reviewers appreciated the attention to an important problem, but raised some substantial concerns about the current manuscript. Based on the reviews, we will not be able to accept this version of the manuscript, but we would be willing to review again a much-revised version. We cannot, of course, promise publication at that time.

If you decide to revise the manuscript for further consideration at PLOS Genetics, please aim to resubmit within the next 60 days, unless it will take extra time to address the concerns of the reviewers, in which case we would appreciate an expected resubmission date by email to plosgenetics@plos.org.

[LINK]

We are sorry that we cannot be more positive about your manuscript at this stage. Please do not hesitate to contact us if you have any concerns or questions.

Yours sincerely,

Diarmaid Hughes

Associate Editor

PLOS Genetics

Josep Casadesús

Section Editor: Prokaryotic Genetics

PLOS Genetics

Reviewer's Responses to Questions

**Comments to the Authors:**

Reviewer #1: This paper uses a FRET approach to interrogate interactions of proteins in the sidewall peptidoglycan (PG) elongosome of Escherichia coli. In particular, the work probes the interactions between the Class B PBP2 transpeptidase (TP) and its SEDS glycosyltransferase (GT) RodA that are mediated by the regulatory proteins MreC and MreD. This study builds on in-cell FRET approaches established and validated by Prof. den Baauwen’s laboratory over the last decade. The work incorporates ideas from structural studies of the MreC-PBP2 interaction and mutant protein constructs discovered as suppressors or null active site variants in previous studies. It also determines the PG crosslinking composition and glycan chain lengths in the mutants. Together, a model emerges for the role of MreC and MreD in elongosome regulation. The model proposes that MreC switches the PBP-RodA activities to an “on” or activated state, whereas MreD returns the functions to the off state. This model is analogous to the one for regulation of septal PG synthesis, which is also mediated by an off/on mechanism.

The sophisticated FRET and PG composition experiments were rigorously performed, and the data are complete with appropriate statistical analyses. The data here are new and will be of considerable importance and interest to the field. This impactful paper is logical and very well written. There are a couple of major points that need to be addressed and/or explained to strengthen certain interpretations and conclusions. These points primarily concern experimental issues that could impact cellular physiology, and hence the interpretation of the results and the working model.

Major points

1. Results and Discussion. Please present data and/or comments about the relative expression levels of the fluorescent-protein constructs relative to wild-type. All of the constructs are expressed from low-copy number plasmids at limiting inducer levels as worked out in previous studies.

Nevertheless, to interpret the physiological relevance of some of these results, it becomes important to know how much combined fluorescent-tag and native untagged protein are being expressed, at least for PBP2, RodA, MreC, and MreD. How close are the protein amounts to the WT levels, and how might increased cellular abundance influence the interpretations? Is the relative 2:1 abundance of MreC to MreD, which is invoked in the Discussion as regulatory, maintained in the constructs? Are the protein mutant constructs expressed at same level as the non-mutant constructs?

It seems like polyclonal antibodies against each of these long-studied E. coli proteins should be available for determinations of relative amounts by quantitative western blotting. At any rate, this topic should be discussed more in the paper, because it affects the interpretations of the FRET experiments and the dominant-negative phenotypes, which can be skewed by relative overexpression.

2. Results and Discussion. L. 348 mentions a severe “tag interference” effect. In the other FRET experiments using two tagged proteins, were any milder tag effects observed that affected cell growth and/or morphology? If so, please indicate these cases and whether other tag effects might affect certain interpretations.

3. L. 209 and L. 443. The method in the paper by Li et. al. to determine absolute proteins synthesis rates is powerful, but makes the huge assumption that all proteins are equally stable, implying that the rates reflect the steady-state amounts. The 2-fold difference in relative amounts of MreC and MreD becomes important later for the model. Has the relative steady-state amounts of these two proteins (or other E. coli elongosome proteins) been determined directly by western blotting for cells grown in the media used? This corroborative information would considerably strengthen some central aspects of the model proposed here. It would also indicate any cleavage of tags off of proteins.

4. L. 243. Was the MreC overexpression phenotype dependent on the tag or was it also observed for overexpression of wild-type MreC? Please clarify.

5. L. 314 and 331. In L. 314, it seems confusing that metcillinam causes a reduction, but active site changes don’t. You might add (“see below”), so this point is not left hanging. In L. 331, is there any structural data for a separate antibiotic binding site? If so, please site.

6. L. 366 and L394. If PBP2(L61R) is activated for TP activity, wouldn’t you expect increased reaction with mecillinam resulting in hypersensitivity? Also, the pbp2(L61R) mutations suppresses delta-mreC mutations and allows growth. That is, the PBP2(l61R) functions fully in the complete absence of MreC, which according to the model is required for stimulation of TP activity. These data do not seem to quite fit with the model. Please clarify.

7. L. 451. The results of Li et al. were obtained for three different growth conditions, which seemed to change the MreC and MreD amounts, although the 2:1 ratio is maintained. It seems that some additional trigger mechanism needs to be added that works in conjunction with MreC/MreD, since their relative amounts are maintained. Please clarify further.

Minor points

8. L. 123. Please change to: “RodA oppositely...”

9. L. 199. For clarity, please change to: “...sufficient for the wild-type interaction...”

10. Table 1, 2, etc. The footnotes might mention that P values for comparisons are shown in the figures.

11. L. 180 and L. 309. Please change to: “As shown”.

12. L. 284. Please delete” “wild type PBP2” (i.e., should read “with MreC and MreD”).

13. L. 285 and elsewhere. “wild-type” as an adjective should be hyphenated.

14. L. 285 and 288. Please change: “Fig. 4d” to “Fig. 4c”.

15. L. 290: Please change the last “dash” between PBP2 and RodA to a “+” because it is referring to the FRET experiment (i.e., “value of RodA+...). Also, please check the labels in the figures where some dashes should be changed to pluses.

16. L. 294. Please consider changing to: “plays a role”.

17. L. 338. Please change to: “If this were the case,”

18. L. 347. Please change: “side” to “site”.

19 L. 432. Please change to: “on PBP2 activity is formally similar”.

20. L. 544. Please change to: “pH 7.2”.

21. L. 566. Please change to: “7000 rpm”.

Reviewer #2: The manuscript presents very solid evidences on interaction between RodA (an integral membrane protein with glycosil transferase activity) and PBP2 (performing transpeptidase activity). Besides, the manuscript presents very interesting data on regulation of the biosynthetic machinery by the MreC and MreD system. Overall the manuscript presents a very nice framework to understand regulation of bacterial cell wall biosynthesis in elongasome. Figures and text are clear, and well supported by the previously reported literature on this topic as well as in agreement with recent structural results.

The manuscript would be of general interest for a broad audience, specially for microbiologists.

Reviewer #3: The manuscript by Liu et al. addresses an important and challenging question in bacterial morphogenesis: how PG synthesis is regulated in the context of the elongasome, the multiprotein complex that orchestrates lateral cell wall growth and maintains rod shape. Understanding the roles of individual components of the elongasome has not been trivial because loss of function mutations are not particularly informative and the system cannot yet be reconstituted in vitro. Liu et al. attempted to bypass these limitations by using FRET to interrogate interactions and potential alternative conformations of elongasome components in vivo. They focused on the interaction between PBP2 and RodA and arrived at a model in which MreC and MreD have antagonistic effects in the regulation of the PBP2-RodA enzymatic core of the elongasome, with MreC being an activator and MreD an inhibitor of PBP2. FRET seems like a good choice to learn more about elongasome regulation, the Den Blaauwen group is experienced with the technique, the manuscript is technically sound and contains plenty of data. However, I am concerned that the main conclusions of the paper are not well supported by the data, as explained below.

Specific comments:

1) The key hypothesis of the paper, that the reduction in PBP2-RodA FRET efficiency produced by MreC reflects an activating conformational change of PBP2, does not seem consistent with other observations. One important inconsistency is with the effects of MreC and MreD overexpression on elongasome function, as inferred from the increase in cell diameter (Fig. 4e). The increase in diameter suggests that both MreC and MreD overexpression make the elongasome less functional. Thus, the FRET reduction produced by MreC overexpression correlates with a less functional elongasome, not the reverse. Likewise, the concomitant overxpression of MreC and MreD rescues both FRET efficiency and elongasome activity, as measured by cell diameter (Figs. 4d and 4e). An important missing experiment in this series is the effect of MreD overexpression alone in PBP2-RodA FRET efficiency. What if FRET efficiency decreases upon MreD overexpression? Even if it does not, the authors should be mindful of alternative explanations for their data, which are probably many, including unexpected ways in which perturbing the stoichiometry of the elongasome will affect its organization.

Another important inconsistency is with the behavior of the L61R mutation. This mutation has no effect on the PBP2-RodA FRET efficiency despite having all the phenotypic and structural features expected of a constitutively active PBP2 protein. This should certainly have prompted more tests of the hypothesis that lower FRET is indeed reporting the PBP2 on state. Instead, the authors take their FRET correlation at face value and create an “ad hoc” explanation for the unexpected behavior of the L61R mutation.

Ideally, the authors should try to provide other types of experimental evidence that tie the reduced FRET with the PBP2 on state. I understand this is not easy, however, the FRET information, as it stands, is just too vague to support the paper´s conclusions.

2) How can the authors conclude that two proteins interact directly from FRET alone? This seems particularly complicated when the efficiencies are low as in the case of the pair MreD-PBP2. What distance does that translate into? 10 nm seems like enough to allow even indirect interactions to produce a FRET signal, in particular when the FP tags are connected to the interacting proteins via flexible linkers, as seems to be the case here.

3) The analysis of the interaction surface between PBP2 and RodA would be strengthened by testing point mutations predicted by Sjodt et al. 2018.

4) Why in the 3 plasmid experiments the FRET efficiency of RodA-PBP2 goes down to 8%, with the empty vector? The authors claim that this is a general effect of the burden of maintaining 3 plasmids but no such reduction is seen with the positive control.

5) The manuscript has many typographical errors, some examples below:

l. 191 – Increase instead of decrease

l. 205 – respectively43?

l. 210 – remove by

l. 222 – remove we

l. 298 – our results have

**Have all data underlying the figures and results presented in the manuscript been provided?**

Reviewer #1: Yes

Reviewer #2: Yes

Reviewer #3: Yes

PLOS authors have the option to publish the peer review history of their article (what does this mean?). If published, this will include your full peer review and any attached files.

Reviewer #1: No

Reviewer #2: Yes: Juan A. Hermoso

Reviewer #3: No

---

## [Decision Letter · Decision Letter 1]

31 Aug 2020

Dear Dr den Blaauwen,

Thank you very much for submitting your Research Article entitled 'MreC and MreD balance the interaction between the elongasome proteins PBP2 and RodA' to PLOS Genetics. Your manuscript was fully evaluated at the editorial level and by independent peer reviewers. The reviewers appreciated the attention to an important problem, but gave widely divergent opinions on its readiness for publication and raised some substantial concerns about the current manuscript. Reviewer #2 asked for the inclusion of additional discussion, including discussion of the implications of a recently published structure. I think adding this discussion would add to the value of the paper and ensure that it is up-to-date. Reviewer #3 strongly requested two additional experiments to test the validity of the model proposed (one experiment to test the effect of MreD over expression on the PBP2-RodA FRET signal; and another experiment to clarify the behaviour of the L61R mutation). Please read carefully Reviewer #3's comments and try to address them. I appreciate that COVID-19 may have affected your ability to carry out normal laboratory work but if your labs are open or opening up I would strongly encourage you to address this reviewers concerns with additional experiments. Based on the reviews, we will not be able to accept this version of the manuscript, but we would be willing to review again a much-revised version. We cannot, of course, promise publication at that time.

If you decide to revise the manuscript for further consideration at PLOS Genetics, please aim to resubmit within the next 60 days, unless it will take extra time to address the concerns of the reviewers, in which case we would appreciate an expected resubmission date by email to plosgenetics@plos.org.

[LINK]

We are sorry that we cannot be more positive about your manuscript at this stage. Please do not hesitate to contact us if you have any concerns or questions.

Yours sincerely,

Diarmaid Hughes

Associate Editor

PLOS Genetics

Josep Casadesús

Section Editor: Prokaryotic Genetics

PLOS Genetics

Reviewer's Responses to Questions

**Comments to the Authors:**

Reviewer #1: The authors have provided thorough, careful, and thoughtful responses to all of the reviewers’ previous comments. Their responses are notable for their deep consideration and detail and address all of the previous issues, within the limits of this experimental system. The result is a highly rigorous study using this form of FRET analysis that leads to an interesting new model for the functions of MreC and MreD in E. coli. My view is that the revised version is an important paper that presents an interesting model that certainly will be further tested by additional approaches in E. coli and other bacterial models.

Reviewer #2: I appreciate authors effort in doing this revised version and I maintain my original point of view about the quality of the present manuscript. I would like, however, indicate some points that would need to be included/discussed in the final version of the manuscript:

1. Concerning the discussion on mecillinam effect on PBP2. There are, at least, two structural precedents of antibiotic binding at allosteric sites far from the active site. The first one is the structure of pneumococcal PBP2X in complex with cefuroxime (Gordon et al, J. Mol. Biol. 2000, 299, 477– 485) and the second one is the structure of PBP2a from MRSA in complex with ceftaroline (Otero et al, Proc. Natl. Acad. Sci. U. S. A. 2013, 110, 16808– 16813). To my knowledge, there is a single example of an antibiotic non-covalently bound to the active site of a PBP, the 3D structure of pneumococcal PBP2X in complex with cefepime (Bernardo-Garcia et al ACS Chem. Biol. 2018, 13, 3, 694–702).

2. In the manuscript (and also in some of the Reviewers comments) there is an important body of discussion about the potential role of PBP2 L61R mutant. During the time for elaboration of this revised version there has been important news, from the structural point of view, about this system. The crystal structure of PBP2 from E. coli (EcPBP2) has been reported alone and in complex with some antibiotics (Levy et al, J. Medicinal Chem. 2019, 62, 9, 4742–4754) and the crystal structure of the PBP2:RodA complex from Thermus thermophilus (TtPBP2:RodA) has been also reported (Sjodt et al, Nature Microbiology 2020, volume 5, pages813–820). It is curious this recent information is not mentioned in the revised manuscript as it deals directly with the discussion and results here presented by a different technique. I think the results here presented should be discussed in the light of the new structural and functional results recently reported for the same complex. The 3D structure of TtPBP2:RodA presents the PBP2 lying onto the RodA protein, very close to the membrane and far from the PG in an inactive configuration for transpeptidation. In this sense, Sjodt et al 2020 propose the interaction with MreC is required in order the PBP2 to reach the PG.

2a. Concerning position of L61 in EcPBP2, I superimposed both structures to know the precise equivalences among E. coli and T. Thermophilus homologues. Indeed, L61 in E. coli corresponds to L43 in T. thermophilus. In the full-length TtPBP2:RodA complex this L43 is located at the interface with RodA in the so-called pedestal domain that Sjodt et al 2020 propose activates glycosyltransferase (GT) activity in RodA. Interestingly, Sjodt et al 2020 have done the equivalent mutant than Liu et al; that is L43R (L61R in E. coli) but some of the conclusions seem to be different: while they confirm the PBP2:RodA complex is formed with the mutant (in agreement with Liu et al) they provide in vitro experimental information that the “GT activity was reduced when Arg was introduced at residue L43”. I think it is opposed to one of the sections by Liu et al manuscript (“PBP2L61R stays in the off state and activates RodA”) (By the way, I am not sure if these data in Sjodt et al 2020 are also in contradiction with that reported by Rohs et al PLOS Gen 2018). This point should be discussed, as maybe with the 3D structure in face, some EPR results could be better understood and/or provide new insights on this matter.

2b. In same sense, the crystal structure of TtPBP2:RodA complex (PDB code 6PL6) revealed an unknown density in between the V-shaped pedestal domain that could correspond to some remnants of MreC. Indeed, creation of a mutant harboring a disulfide bridge linking both sides of the PBP2 pedestal domain (thus closing the central cavity) resulted in severe morphological defects, probably as a result of a reduced interaction with MreC. How FRET results agree with these data? Could you interpret MreC interaction with PBP2 in the light of this site? Are there other potential explanations?

Reviewer #3: The revised version of the manuscript by Liu et al. has failed to address the important inconsistencies pointed out in my review that call into question the validity of their main conclusions.

The authors opted not to carry out an obvious and important control – the effect of MreD overexpression on the PBP2-RodA FRET signal. The goal of this control is to rule out the possibility that reduction in the PBP2-RodA FRET signal observed with MreC could be a general response to perturbing the stoichiometry of the Rod complex. The authors argued that “Because of the lack of strong effects on the morphology of the cells by MreD, we do not expect an effect of MreD on the interaction between RodA and PBP2”. By the same logic, we would not expect MreC to have an effect on the PBP2-RodA interaction either.

The second important point that was not addressed in any substantial way is the unexpected FRET behavior of the PBP2L61R mutant. This is a mutant that activates the Rod system in the absence of MreC, thus strongly suggesting that it makes the PBP2-RodA complex adopt the same conformation induced by MreC. Thus, the FRET behaviour of this mutant should mimic the FRET behavior in response to MreC. If this expectation is not borne out by the data I would be very worried about the validity of my hypothesis (low FRET = active conformation). Instead, the authors insist in explaining this unexpected result with convoluted and improbable reasoning. They argue that the PBP2L61R-RodA complex conformation need not be the same as the MreC-activated PBP2-RodA conformation because the mutation only stimulates the glycosyltransferase and not the transpeptidase activity of the complex. This seems to be based on the assumption that MreC stimulates both the glycosyltransferase and transpeptidase of PBP2-RodA, but, as far as I can tell, there is no data to support this. In fact, the observation that mutations in RodA can also bypass MreC led to the model that Rod complex activation occurs primarily through activation of the glycosyltransferase activity of RodA (Rohs et al., 2018). The recently published structure of the PBP2-RodA complex seems to also support this model (Sjodt et al., 2020).

As stated in my initial review, I suspect it will not be simple to sort out what is going on with PBP2L61R. However, there are relatively simple experiments that could have been be done to increase our confidence that the FRET data is really telling us what the authors claim. In addition to the MreD control, it would be possible to test mutations that prevent the interaction between PBP2 and MreC (Contreras-Martel et al., 2017) as well as mutations that prevent the allosteric activation of RodA by PBP2 (Sjodt et al., 2020). If these mutants behave as expected (being insenstive to MreC in a 3 plasmid experient) this would substantially strengthen the conclusions of the manuscript.

Minor points:

l 253. In contrast to what is stated, Supplementary Figure 5b shows that the co-overexpression of MreC and D leads to the same aberrant cell shape as the overexpression of the individual proteins.

l 268. ... likely attributed ...

**Have all data underlying the figures and results presented in the manuscript been provided?**

Reviewer #1: Yes

Reviewer #2: Yes

Reviewer #3: Yes

PLOS authors have the option to publish the peer review history of their article (what does this mean?). If published, this will include your full peer review and any attached files.

Reviewer #1: No

Reviewer #2: No

Reviewer #3: No

---

## [Decision Letter · Decision Letter 2]

12 Nov 2020

Dear Dr den Blaauwen,

We are pleased to inform you that your manuscript entitled "MreC and MreD balance the interaction between the elongasome proteins PBP2 and RodA" has been editorially accepted for publication in PLOS Genetics. Congratulations!

Yours sincerely,

Diarmaid Hughes

Associate Editor

PLOS Genetics

Josep Casadesús

Section Editor: Prokaryotic Genetics

PLOS Genetics

Comments from the reviewers (if applicable):

Reviewer's Responses to Questions

**Comments to the Authors:**

Reviewer #1: This revised paper is an important contribution to the field of bacterial peptidoglycan synthesis for several reasons. Foremost, the authors use FRET to test in vivo several major hypotheses about the interactions and regulation of the peptidoglycan elongasome in Gram-negative Escherichia coli. These hypotheses have emerged from recent independent genetic, structural, and biochemical studies, but they have never really been drawn together and tested as an intact system in the cell. Of note, this study is one of the first to focus on possible functions of the integral-membrane MreD protein, which has often been difficult to tag for detection and study biochemically. The paper supports the idea that the MreC regulator changes the interaction between Class B PBP2 and RodA, thereby turning on this peptidoglycan synthase for elongation. The paper provides the new observation that MreD seems to antagonize this MreC activation, returning the PBP2:RodA synthase to an off state. Thus, there seems to be regulation by partner switching of MreC or MreD interacting with PBP2 in the PBP2:RodA complex. The paper makes the interesting analogy to the on-off mechanism that regulates septal peptidoglycan synthesis, mediated by FtsBLQ (off) and FtsN (on), suggesting a general paradigm for peptidoglycan synthesis in E. coli. What throws the switch between on and off for elongation and the role of MreB organization are interesting problems for the future.

The paper includes several other new conclusions arrived at by clever heterologous domain-swap FRET experiments, including that the small cytoplasmic domain of PBP2 is important for interactions with the MreCD proteins. In addition, the paper provides the unexpected results that mecillinam affects the interaction between PBP2 and RodA, independent of PBP2 catalytic activity, suggesting the existence of an allosteric site of mecillinam binding in PBP2 in vivo. The paper also changes the interpretation of the effect of the PBP2(L61R) amino acid change, which was initially proposed to keep the PBP2(L61R) protein in the “on” state in the absence of MreC. However, within the experimental limits of this FRET approach, the PBP2(L61R) mutant protein does not alter the interaction with RodA compared to WT PBP2. By determining peptidoglycan crosslinking levels and glycan chain lengths, the paper shows that the glycan chain lengths are dramatically increased in the PBP2(L61R) mutant, resulting in defective peptidoglycan that could underly changes in cell shape and in antibiotic sensitivity. Together, the results in this paper strongly suggest that the PBP2(L61R) mutant protein stimulates the glycosyltransferase activity of RodA, instead of the transpeptidase activity of PBP2.

This paper contains a vast amount of high-quality, well-presented data from challenging experimental methods. Over the rounds of reviews, the authors have added additional data and carefully addressed each of the reviewers’ comments. The current version is a highly refined version compared to the starting one. This last version is an interesting and important contribution to the field that will stimulate further tests of these interactions and mechanisms both in vivo and in vivo.

Reviewer #2: In this revised version authors have properly answered all my previous comments and suggestions. I thanks the authors for the effort in preparation of this new version of the manuscript.

Reviewer #3: The second revision of the Liu et al. manuscript is substantially improved. The addition of the MreD control has eliminated an important concern about the specificity of the MreC FRET effect. This, in turn, increases our confidence that the unexpected behavior of the L61R mutant could indeed indicate a different “activation state” than the one brought about by MreC. The authors also did a good job at discussing the results of the L61R mutant in a more balanced and objective way.

I still wish the authors were a little more cautious in their interpretation of the FRET results. A case in point is the meaning of FRET reduction, which sometimes is interpreted as a conformational change (MreC effect on PBP2-RodA) and sometimes as a loss of interaction (MreD effect on PBP2-MreC). This makes sense in light of their model, but without other types of supporting evidence, it is far from being the only explanation for their results. It is worth nothing that there is at least one result that is inconsistent with the idea that MreC and MreD have some sort of mutually exclusive interaction with PBP2-RodA: in the experiments with the domain swapped mutant MalFNTPBP2, MreD no longer antagonizes MreC despite still interacting with MalFNTPBP2 (Fig. 4c). This is not a major problem – the manuscripts conclusions and model are overall well supported and important - but it is a reminder that things are often more complicated than we´d like them to be.

**Have all data underlying the figures and results presented in the manuscript been provided?**

Reviewer #1: Yes

Reviewer #2: Yes

Reviewer #3: Yes

PLOS authors have the option to publish the peer review history of their article (what does this mean?). If published, this will include your full peer review and any attached files.

Reviewer #1: No

Reviewer #2: No

Reviewer #3: No

**Data Deposition**

http://datadryad.org/submit?journalID=pgenetics&manu=PGENETICS-D-19-01848R2

**Press Queries**

---

## [Editor Report · Acceptance letter]

22 Dec 2020

PGENETICS-D-19-01848R2 

MreC and MreD balance the interaction between the elongasome proteins PBP2 and RodA 

Dear Dr den Blaauwen, 

We are pleased to inform you that your manuscript entitled "MreC and MreD balance the interaction between the elongasome proteins PBP2 and RodA" has been formally accepted for publication in PLOS Genetics! Your manuscript is now with our production department and you will be notified of the publication date in due course.

With kind regards,

Livia Horvath

PLOS Genetics

On behalf of:
